# The transcription factor PREP1(PKNOX1) regulates nuclear stiffness, the expression of LINC complex proteins and mechanotransduction

Divya Purushothaman [1✉], Laura F. Bianchi [1], Dmitry Penkov[1,2], Alessandro Poli[1], Qingsen Li[1], Jelena Vermezovic[1], Francesca M. Pramotton [3,4], Ramveer Choudhary[1], Fabrizio A. Pennacchio[1], Elena Sommariva [5], Marco Foiani [1], Nils Gauthier[1], Paolo Maiuri [1] & Francesco Blasi [1✉]

Mechanosignaling, initiated by extracellular forces and propagated through the intracellular cytoskeletal network, triggers signaling cascades employed in processes as embryogenesis, tissue maintenance and disease development. While signal transduction by transcription factors occurs downstream of cellular mechanosensing, little is known about the cell intrinsic mechanisms that can regulate mechanosignaling. Here we show that transcription factor PREP1 (PKNOX1) regulates the stiffness of the nucleus, the expression of LINC complex proteins and mechanotransduction of YAP-TAZ. PREP1 depletion upsets the nuclear membrane protein stoichiometry and renders nuclei soft. Intriguingly, these cells display fortified actomyosin network with bigger focal adhesion complexes resulting in greater traction forces at the substratum. Despite the high traction, YAP-TAZ translocation is impaired indicating disrupted mechanotransduction. Our data demonstrate mechanosignaling upstream of YAP-TAZ and suggest the existence of a transcriptional mechanism actively regulating nuclear membrane homeostasis and signal transduction through the active engagement/disengagement of the cell from the extracellular matrix.

[1] IFOM (Foundation FIRC Institute of Molecular Oncology), via Adamello 16, 20139 Milan, Italy. [2] National Medical Research Center of Cardiology, 3rd Cherepkovskaya Street, 15a, 121552 Moscow, Russia. [3] EMPA-Materials Science and Technology, 8600 Dubenforf, Switzerland. [4] Institute for Mechanical Systems, ETH, 8092 Zurich, Switzerland. [5] Unit of Vascular Biology and Regenerative Medicine, Centro Cardiologico Monzino, Via Carlo Parea, 4, 20138 Milan, Italy. ✉email: divya.purushothaman@ifom.eu; Francesco.blasi@ifom.eu

PREP1 is a homeodomain transcription factor of the TALE family, essential for embryonic development[1]. Mice lacking PREP1 fail to gastrulate resulting in embryonic lethality at e6.5 due to DNA damage-dependent apoptosis, while mouse embryonic stem cells lacking PREP1 fail to differentiate[2]. In the embryo, most PREP1 target genes are not strictly developmental but rather related to basic cellular functions such as cell adhesion, histone modification and signal transduction[3–6]. Here we describe a role for PREP1 in the process of mechanosignaling which could influence the differentiation potential of a cell and impact many biological functions.

During mechanosignaling, extracellular matrix rigidity and coordinated signal transduction pathways, initiate and finetune the lineage specifications of multipotent precursors[7,8]. In a cell, focal adhesions act as the major hub of multiple mechanosensors like vinculin, Src, Fak, p130CAS etc which transmit the extracellular forces to the cell interior through a contractile actomyosin network. LINC (Linker of Nucleoskeleton and Cytoskeleton) complex proteins (SUN1/2 and NESPRINS) which connect the cytoskeleton to the nucleus, close the circuit, providing a continuous route for the propagation of extracellular cues to the nucleus. Nucleus shuttling proteins like β-catenin, c-Abl, zyxin and mechanotransducer YAP-TAZ act to transduce the signals conveyed by focal adhesion pathway[9]. Thus, extracellular signals activate a variety of signaling pathways which culminate in the nucleus and trigger gene transcription[9]. However, the cell is not just a passive receiver or responder of mechanical forces exerted on it. It can actively remodel the actomyosin cytoskeleton in response to extracellular stress by a process called mechanoreciprocity[10]. Here we describe a cellular mechanism mediated by transcription factor PREP1 that regulates nucleo-cytoskeletal coupling and mechanoreciprocity of the cells.

Previously we showed that PREP1 decrease leads to an anomalous DNA replication timing which affects mostly the nuclear membrane-bound DNA[11]. Exploring these results, we focused our attention on the structure of the nucleus upon PREP1 depletion. Here, we show that PREP1 binds to promoter regions of inner nuclear membrane proteins SUN1, SUN2 and LAP2 genes and upon reduction of PREP1 the expression of these genes and proteins is affected. Indeed, PREP1 depletion leads to softer nuclei, a decrease in SUN2 and the concomitant increase of SUN1 and LAP2. In parallel, in the cytoplasm these cells display strengthened actomyosin bundles terminating in bigger focal adhesion complexes exerting increased traction force on the substratum. However, YAP-TAZ nuclear translocation is reduced and unresponsive to changes in substrate rigidity, revealing disrupted mechanotransduction. Our study uncovers a hitherto unknown transcriptional regulation of the cellular mechanosignaling pathway.

## Results

### PREP1 depletion perturbs nuclear envelope composition and mechanical properties

We used the bone osteosarcoma cell line U2OS and the cervical cancer cell line HeLa as our experimental system to study the effects of PREP1 knockdown in the nucleus. Cells transfected with a cocktail of siRNA oligos against PREP1[11] showed efficient downregulation of the protein at 48 h post transfection (Supplementary Fig. 1a, Fig. 1e, f). An siRNA against Luciferase gene (siLUC) was used as control.

In U2OS cells, immunofluorescence analysis using an anti-LAMIN B1 antibody and DAPI staining to mark the nuclear envelope and nucleus respectively, revealed severe nuclear deformation in PREP1 depleted compared to control cells (Fig. 1a). Approximately 50–60% of the cells showed evident nuclear deformations upon PREP1 depletion (Fig. 1b). Using a custom-made macro, we analyzed and quantified the nuclear deformations in PREP1 downregulated cells using parameters such as lamin fragmentation index (severity of nuclear invaginations), width to length ratio and roundness, (see methods): siPREP1 cells showed significant differences in the parameters tested (Supplementary Fig. 1a). As nuclear deformations are usually associated with mechanical defects of the nuclear envelope[12–14], we measured the stiffness of these cells (Fig. 1c) using Atomic Force Microscopy (AFM)[15]. PREP1 depleted cells had lower elastic properties (Youngs' modulus) indicating that PREP1 depletion decreases nuclear stiffness and compromises nuclear morphology (Fig. 1a, c).

Abnormal nuclear morphology and mechanics often results from aberrations in the nuclear lamina, the interconnecting meshwork of intermediate filaments composed of LAMIN B and LAMIN A/C which provide structural support to the lipid bilayer of nuclear membrane[16–18]. Also, several lamina interacting inner nuclear membrane proteins such as EMERIN, SUN1/2, Nesprins and LBR are involved directly or through their interaction with Lamins, in the formation and maintenance of proper nuclear and chromatin architecture and nuclear anchoring[12,18–23]. Hence, we methodically analyzed the protein expression by immunoblotting and their localization by immunofluorescence in PREP1 depleted cells. Quantification of nuclear intensity of each protein in IF was done using custom-made macro in ImageJ. We first looked at LAMIN A and LAMIN B. While we did not see substantial differences in the localization of LAMIN B, nuclear localization of LAMIN A/C was consistently lower in PREP1 depleted U2OS cells (Fig. 1d). However, total protein levels did not show any change in immunoblots (Fig. 1e). This prompted us to examine the expression and localization of other lamina associated nuclear envelope proteins to understand their status upon PREP1 depletion. Immunofluorescence analysis revealed an increase in the nuclear signal intensity of SUN1 and a concomitant decrease in the intensity of SUN2 nuclear signal in PREP1 depleted U2OS cells (Supplementary Fig. 1b). KASH domain protein NESPRIN2-2 also showed increased intensity at the nuclear envelope while LBR and NESPRIN1 were not affected (Supplementary Fig. 1b).

Further, we checked whether these changes can be attributed to changes in protein expression by immunoblotting of total cell lysates. Both SUN1 and SUN2 showed substantial changes at the protein level (Fig. 1f). Confirming the IF data, SUN1 was increased while SUN2 was decreased in PREP1 depleted cells. At the same time, LBR, EMERIN or NESPRIN1 were not changed. We detected an increase in the beta isoform of LAP2 protein both by IF and immunoblot analysis (Supplementary Fig. 1c & Fig. 1f), however the significance of this isoform specific increase is not pursued in the present study. Our data shows that depletion of PREP1 results in specific changes in SUN domain proteins and an overall perturbation of the lamina and associated proteins in U2OS cells.

We then moved to HeLa cells and repeated the analysis to check the universality of our results. PREP1 depleted HeLa cells showed similar trends in the expression of SUN1, SUN2, and LAP2 in immunofluorescence and immunoblotting assays confirming that the regulation of these proteins by PREP1 is not cell line specific (Supplementary Fig. 1d, e). Additionally, in HeLa cells, the changes in nuclear envelope proteins were accompanied by decrease in nuclear stiffness similar to that of PREP1 depleted U2OS cells (Supplementary Fig. 1f), confirming that PREP1 influences the expression of nuclear envelope proteins and nuclear mechanics. Intriguingly, however, HeLa nuclei did not present the nuclear deformations (Supplementary Fig. 1d) characteristic of U2OS PREP1 depleted cells even though the disruption of nuclear membrane stoichiometry was comparable and nuclear mechanics impaired (Supplementary Fig. 1d, e, f).

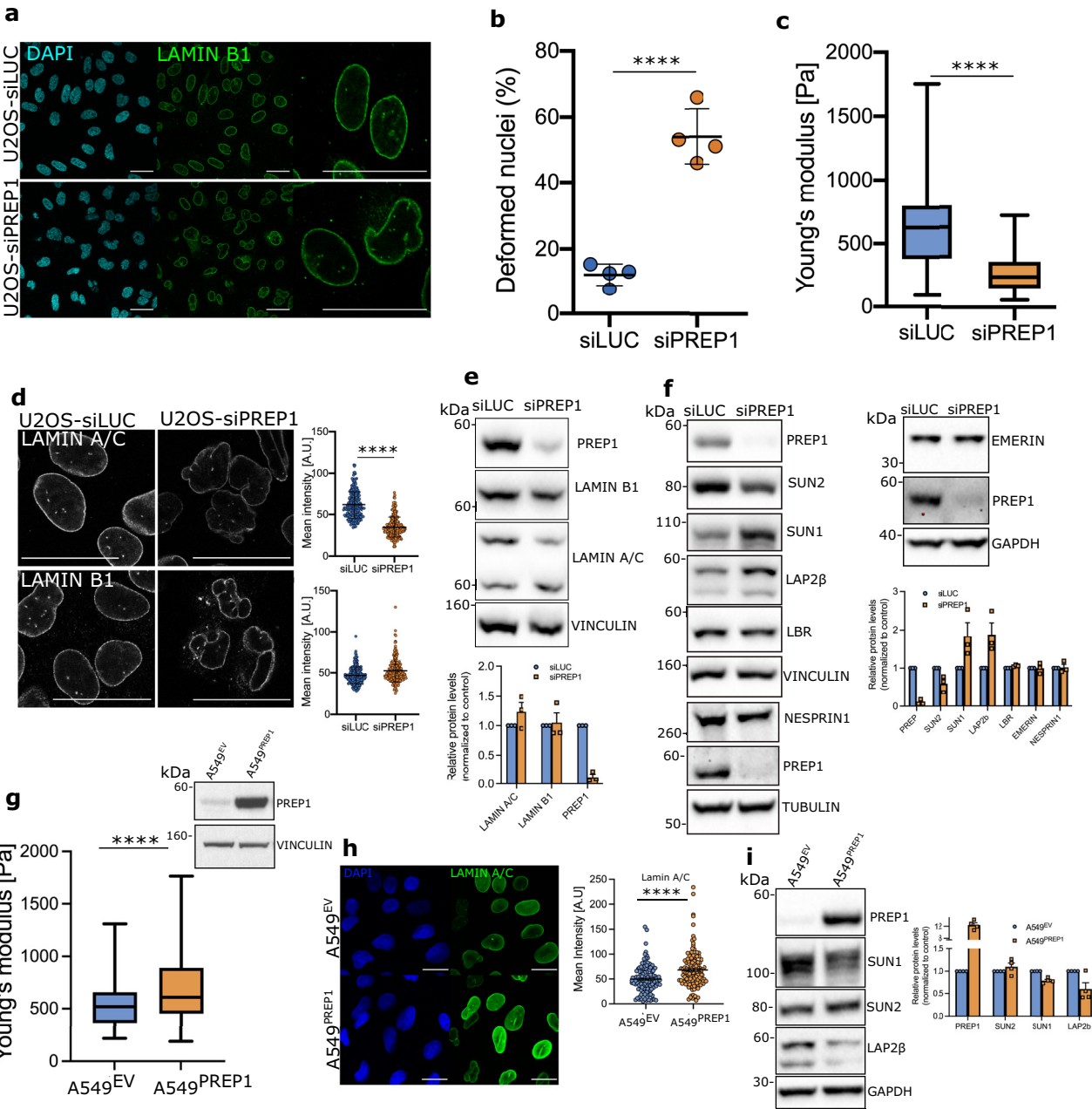

**Fig. 1 PREP1 regulates nuclear envelope composition and nuclear rigidity. a** Confocal images of control (siLUC) and PREP1 depleted (siPREP1) U2OS cells. DAPI staining (blue) detects the nucleus while LAMIN B1 staining (green) detects the nuclear lamina. Right panel shows nuclei at higher magnification in siLUC and siPREP1 cells. Representative images of >3 independent experiments are shown. Scale bar is 40 μm. **b** Scatter dot plot showing percent nuclear deformation in control (siLUC) and PREP1 depleted (siPREP1) U2OS cells in four independent experiments. $N = 600$ for siLUC and 884 for siPREP1. Error bars denote mean ± SD of four independent experiments. $p < 0.0001$. **c** Box plot showing Youngs' modulus measured by AFM in control (siLUC) and PREP1 depleted (siPREP1) U2OS cells. $N = 40$ cells. $p < 0.0001$. **d** Immunofluorescence images of control (siLUC) and PREP1 depleted (siPREP1) cells showing LAMIN A/C (upper panel) and LAMIN B1 (lower panel). Scale bar is 40 μm. Corresponding scatter plots show quantification of fluorescence intensity of LAMIN A/C (upper panel) and LAMIN B1 (lower panel). $N = 232$ (siLUC) and 244 (siPREP1) for LAMIN B1 analysis and $N = 243$ (siLUC) and 266 (siPREP1) for LAMIN A/C analysis. **e** Representative immunoblots of control (siLUC) and PREP1 depleted (siPREP1) U2OS cells showing LAMIN A/C and LAMIN B1 protein levels. Data is representative of three independent experiments. Bar graph shows densitometric analysis of LAMIN A/C and LAMIN B1 protein in three independent experiments. **f** Representative immunoblots of control (siLUC) and PREP1 depleted (siPREP1) U2OS cells showing various nuclear envelope proteins. Bar graph shows densitometric analysis of the proteins from three or more independent experiments. **g** Box plot showing Youngs' modulus measured by AFM in empty vector (A549[EV]) and PREP1 overexpressing (A549[PREP1]) A549 cells. $N = 40$ cells. Inset immunoblot shows the PREP1 protein in A549[EV] and A549[PREP1] cells. **h** Immunofluorescence images showing LAMIN A/C localization in A549 cells expressing empty vector (A549[EV]) or PREP1 (A549[PREP1]). Scale bar is 20 μm. Scatter plot shows mean fluorescence intensity of LAMIN A/C of A549[EV] or A549[PREP1] cells. $N = 125$ cells. Data is representative of four independent experiments. $p < 0.0001$. **i** Immunoblot showing SUN1, SUN2, and LAP2β in A549[EV] and A549[PREP1] cells. Bar graph shows densitometric analysis of the protein intensity from three independent experiments. Error bars denote mean ± SE.

This suggested that changes in SUN1, SUN2, and LAP2 were not sufficient to trigger nuclear deformations. Perhaps, a nuclear independent event is necessary for the deformation of the nucleus in U2OS cells: this is explored below. Altogether, our data show that the absence of PREP1 upsets the stoichiometric balance of nuclear membrane associated proteins SUN1, SUN2, and LAP2 and causes nuclear softness.

To confirm that the nuclear phenotypes were indeed due to the reduction of PREP1, we tried to perform rescue experiments with PREP1 in U2OS cells. However, transfections with vectors overexpressing PREP1, induced massive cell death, independently of the transfection mechanism. It has been reported that PREP1 overexpression in fibroblasts induces apoptosis[24] that might explain the reason for massive cell death in U2OS cells upon PREP1 overexpression. We, therefore, turned to the human lung cancer cell line A549 in which retroviral expression of PREP1 yielded stable cell lines with no substantial loss in cell viability[25]. Thus, we were able to produce stable cell lines with high levels of PREP1. The nuclear stiffness of A549[PREP1] cells was analyzed by AFM. The results showed that the overexpression of PREP1 increased the Youngs' modulus, indicating higher stiffness than control cells (Fig. 1g). Further, immunostaining revealed an increase in LAMIN A/C compared to empty vector control (Fig. 1h). These cells also showed a slight increase in the total SUN2 levels and a decrease in SUN1 and LAP2β levels compared to control cells in immunoblotting (Fig. 1i). Thus, our data show that overexpression of PREP1 reverts many of the phenotypes observed in PREP1 depletion. The phenotype rescue we see upon PREP1 overexpression in an altogether different cell line confirms that changes in nuclear membrane protein stoichiometry and nuclear stiffness is indeed mediated by PREP1 in a variety of cells.

**PREP1 transcriptionally regulates inner nuclear membrane proteins SUN1, SUN2, and LAP2.** It was established previously that PREP1 binds to both enhancers and promoters of a variety of genes[3]. Given that PREP1 depletion led to changes in the protein levels of SUN1 and SUN2, we considered the possibility that PREP1 is a transcriptional regulator of these genes. Thus, we estimated by qPCR analysis, the transcript levels of SUN domain proteins in the absence of PREP1. Interestingly, *SUN2* mRNA was downregulated while *SUN1* and *LAP2* transcripts were significantly upregulated in both PREP1 depleted HeLa and U2OS cells (Fig. 2a, b). Further confirming the immunoblot data, no difference in transcript levels were seen in *LAMIN A/C*, and *LBR* (Fig. 2a, b).

ChIP-seq analysis of PREP1 binding in HeLa cells[11] (GSE101776) showed that it binds very strongly to the proximal promoter of *SUN2* and moderately to a regulatory element of *LAP2* (*TMPO*) gene. Bowtie analysis shows that this last element bears characteristic features of enhancer (DNAse hypersensitivity, P300[+], H3K27Ac[+]) and is located outside of the gene about 15 kbs from TSS. No binding was recorded in the vicinity of *SUN1* gene in spite of many H3K27Ac[+] regions located both outside and inside the gene. PREP1 binding to *SUN2* promoter and *TMPO* enhancer was confirmed by ChIP-PCR in HeLa cells (Supplementary Fig. 2a). Both binding sites contain decameric sequences (TGACTGACAG and TGATGGACAG respectively), the most common PREP1 binding consensus sequence[3]. Even if no binding of PREP1 was detected at *SUN1*, two octameric sequences (TGATGGAC, TGAGTAAT), other possible PREP1 binding sites[3], are located inside the gene around 2 kbs from the TSS[11].

To test if these DNA binding sites are present in other cell types, we analyzed PREP1 binding profile in primary human cardiac Mesenchymal Stromal Cells (MSCs) by Chip-seq. The data is deposited at GEO and is retrievable under the GSE160286 accession number. In both cases there was PREP1 binding near *SUN2* and *TMPO* genes at the same positions as in HeLa (Fig. 2c). Contrary to HeLa cells, in MSCs, *SUN1* regulatory element was bound by PREP1 confirming that also *SUN1* can be a direct target gene. It is possible that difference in the chromatin conformation state between pluripotent (MSC) and differentiated (HeLa) cells is responsible for the difference in binding. Interestingly, we found that PREP1 binding to these genes is conserved across species, as *Sun2* and *Sun1* was strongly bound by Prep1 in the promoter and proximal promoter region respectively (Supplementary Fig. 2b) of mouse embryonic stem cells[4] (GSE63282) and cells derived from e10.5 mouse embryo trunk[3] (GSE39609). *Tmpo(Lap2)* gene was not bound by Prep1 in the mouse cells investigated. Overall, our data suggest that all three genes (*SUN1, SUN2, TMPO/LAP2*) are direct targets of PREP1 in a cell- and tissue-specific manner. We propose that PREP1 is able to bind *SUN1, SUN2* and *TMPO (LAP2)* genes and regulates their expression.

Our data show that PREP1 regulates the nuclear membrane composition of SUN1, SUN2 and LAP2 transcriptionally and affects the nuclear stiffness. Next, we dissected the apparent discrepancy in the nuclear deformation phenotype between U2OS and HeLa cell lines. To begin with, we tested the individual contribution of SUN1/2 to the nuclear deformation phenotype in U2OS cells. Interestingly, SUN2 depletion had no effect on nuclear morphology in U2OS cells (Fig. 3a), arguing against a role for SUN2 alone in the nuclear deformation induced by PREP1 depletion. However, SUN2 depleted U2OS cells demonstrated softer nuclei as measured by AFM analysis (Supplementary Fig. 3a) even with morphologically normal nuclei suggesting that SUN2 depletion could contribute to part of the phenotypes observed upon PREP1 depletion. SUN1 accumulation is known to contribute to the abnormal nuclear phenotypes displayed by HGPS (Hutchinson-Gilford Progeria Syndrome) fibroblasts and its reduction rescues the nuclear defects[23]. To test if nuclear deformation in PREP1 depleted U2OS cells is due to SUN1 accumulation, we downregulated SUN1 using siRNA and assessed the nuclear deformation. However, SUN1 depletion not only failed to correct the nuclear deformation seen in PREP1 deficient cells but also caused nuclear deformations on its own (Fig. 3b). All these data taken together suggest the involvement of other factors, possibly nuclear extrinsic factors, in the defective nuclear morphology observed in PREP1 depleted U2OS cells. These data also suggested that nuclear phenotype observed upon PREP1 downregulation can not be attributed to the loss or gain of individual nuclear envelope proteins but rather to a cumulative effect of the changes.

**Reinforced actomyosin network causes nuclear deformation in PREP1 depleted cells.** The LINC complex proteins, SUN1/2 are connected to the actin cytoskeleton[26]. Actin based nuclear confinement and actomyosin contractility are implicated in nuclear envelope rupture of interphase nuclei and are known to influence nuclear dysmorphia in cancer cells[27,28]. Hence, we looked at the status of actin stress fibers in PREP1 depleted U2OS and HeLa cells by phalloidin staining. To our surprise, both HeLa and U2OS cells displayed increased actin stress fiber formation upon PREP1 depletion (Fig. 3c and Supplementary Fig. 3b). Four different categories of actin stress fibers have been identified depending on the localization and assembly: dorsal and ventral stress fibers, transverse arcs and perinuclear actin cap[29,30]. Of these, the perinuclear actin cap, formed by actin stress fibers positioned over the nuclei is able to interact with LINC complexes and thereby regulate the shape of interphase nuclei[30]. To discriminate the perinuclear actin structures from dorso-ventral

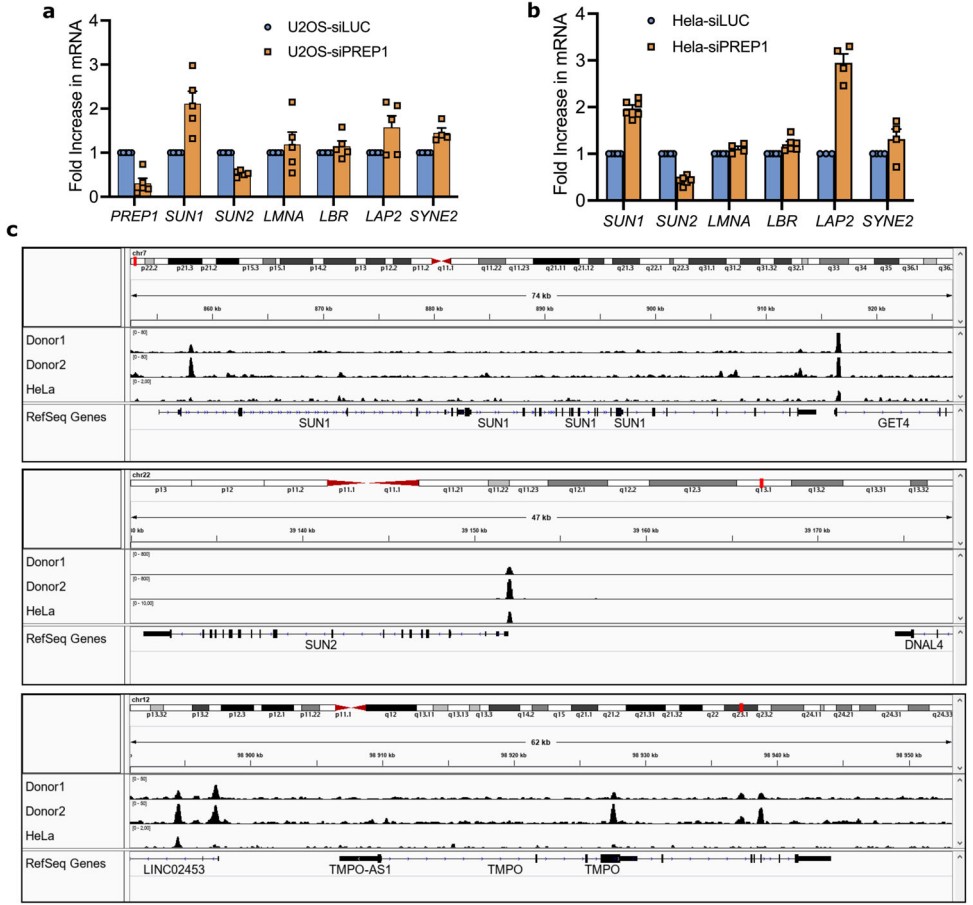

**Fig. 2 Transcriptional regulation of *SUN1, SUN2* and *LAP2* by PREP1. a, b** Bar graph shows transcript levels of SUN1, SUN2, LMNA, LBR, LAP2 and SYNE2 in control (siLUC) and PREP1 depleted (siPREP1) U2OS cells (**a**) or HeLa cells (**b**) assessed by qPCR. Error bars show mean ± SEM of three or more independent experiments. **c** PREP1 binding profile (Big Wig files) to the loci of human *SUN1* (upper panel), *SUN2* (middle panel) and *TMPO/LAP2* (bottom panel) genes in the IGV genome browser are shown. The top bar shows the entire human chromosome 7 (upper panel), human chromosome 22 (middle panel) or human chromosome 12 (bottom panel) with the various chromosomal positions (p and q). The lower corresponding rows show the binding profiles of PREP1 on *SUN1* (upper panel), *SUN2* (middle panel), or *TMPO* (*LAP2*, bottom panel) in Chip-seq experiments using cardiac mesenchymal stromal cells from two human donors and HeLa cells.

stress fibers, we imaged the phalloidin stained siLUC and siPREP1 U2OS and HeLa cells along the *z*-axis using confocal microscopy. Basal and apical stacks along the *z*-axis were projected separately to visualize the actin structures (Fig. 3d). This analysis demonstrated the presence of comparably thicker actin bundles resembling basal/ventral stress fibers in PREP1 depleted HeLa and U2OS cells with respect to control cells (Fig. 3d, lower panel). Intriguingly, only U2OS but not HeLa cells revealed the presence of structures resembling perinuclear actin caps upon PREP1 depletion (Fig. 3d, upper panel).

To visualize perinuclear actin cap more clearly, we grew HeLa and U2OS cells on fibronectin-coated micropatterns which promotes the actin cap formation[30]. As expected, perinuclear actin caps were absent in control HeLa and U2OS cells. However, almost 80% of PREP1 depleted U2OS cells showed the buildup of perinuclear actin caps when grown on micropatterns (Fig. 3e). Interestingly, PREP1 depletion did not cause perinuclear actin cap formation in HeLa cells grown on micropatterns (Supplementary Fig. 3c) suggesting that the apparent normal nuclear morphology in HeLa cells could be due to the lack of cytoplasmic pulling forces exerted through the apical actomyosin bundles connected to the nucleus through LINC complex. Henceforth, we asked whether the pharmacological inhibition of actomyosin contractility would rescue the nuclear deformation phenotype in

U2OS cells. Treatment with 40 μM blebbistatin, a myosin ATPase inhibitor, for 20 min disrupted actomyosin bundle formation and restored normal nuclear morphology to a significant extent in PREP1 depleted U2OS cells (Fig. 3f). While, the rescue is not complete, we were able to observe reversal of parameters such as lamin fragmentation index and roundness in the siPREP1 cells treated with blebbistatin (Supplementary Fig. 3d). Thus, coupling the increased softness of the nucleus with a higher contractility of the reinforced perinuclear actin cap is responsible for the nuclear deformability observed in PREP1 depleted U2OS cells. In the absence of external stressors as in HeLa, nuclei remain apparently normal.

**PREP1 depletion perturbs the focal adhesion pathway in the cytoplasm.** The actomyosin bundles terminate in focal adhesions, multiprotein complex structures at the cell membrane which act as the primary mechanosensors of the cell[31,32]. The tension generated by the actomyosin network is directly linked to the focal adhesion maturation and is the conduit for signal transmission from extracellular matrix to the nucleus during mechanosignaling[33]. Hence, we directly assessed the size of focal adhesion complexes using vinculin staining in PREP1 depleted HeLa and U2OS cells. Expectedly, PREP1 depletion led to the formation of larger adhesion foci in HeLa and U2OS cells (Fig. 4a, b, Supplementary Fig. 4a, b). This data

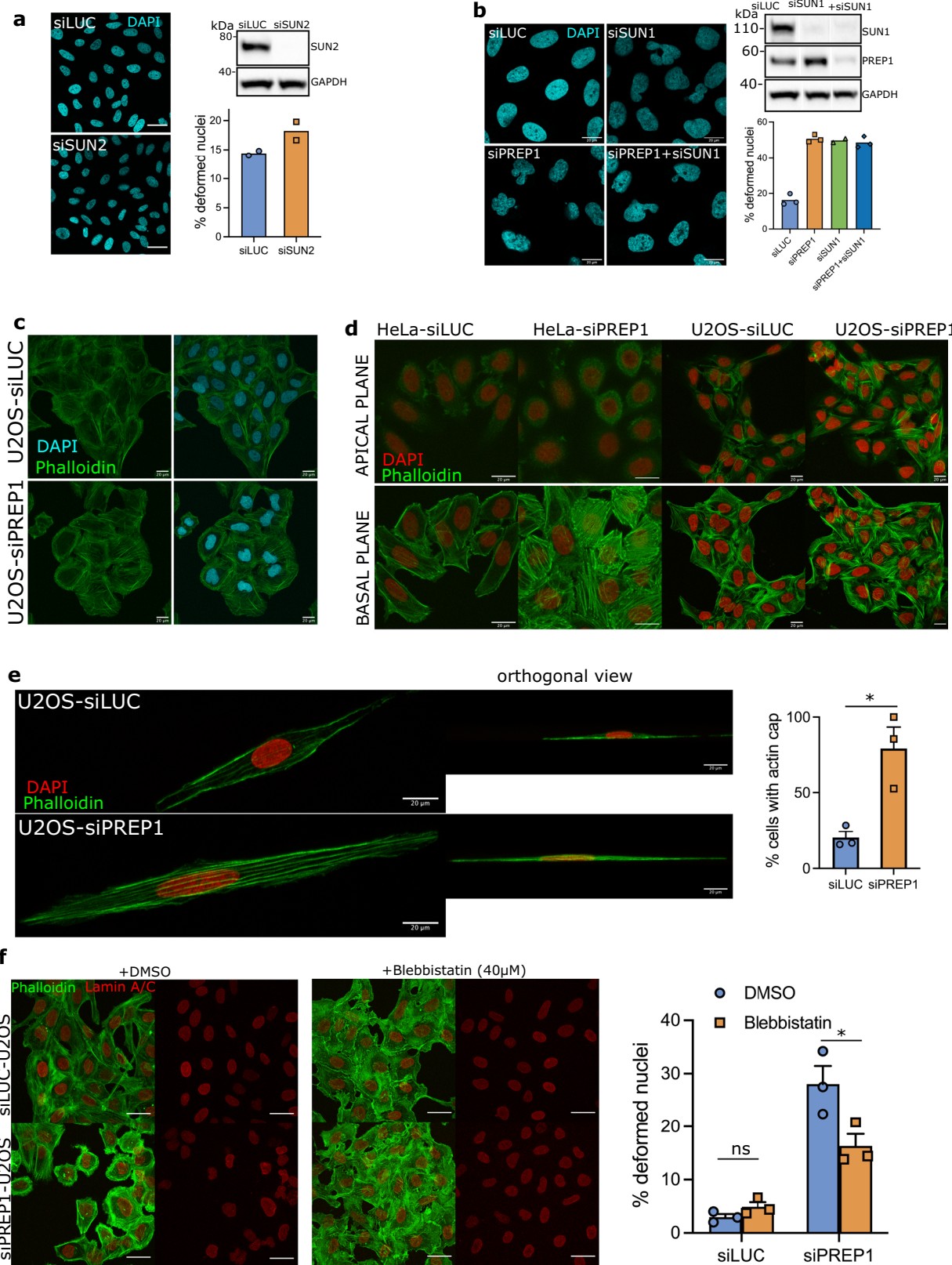

is in conformity with the published reports of close correlation between the size of the focal adhesions and the force transmitted through actomyosin contractility[34]. In line with this, traction force microscopy measurements confirmed that the mean traction force exerted by PREP1 depleted U2OS cells on the substratum is higher

than in control cells (Fig. 4c). Similar results were observed in HeLa cells depleted of PREP1 (Supplementary Fig. 4c).

Disruption of LINC complexes affects perinuclear/apical actin structure[35–37]. To understand if the heavy formation of actin network is caused by a disrupted LINC complex, we

**Fig. 3 PREP1 depletion induces actin stress fiber formation. a** Confocal images of DAPI (nucleus, blue) staining in control (siLUC) and SUN2 depleted (siSUN2) U2OS cells. Scale bar is 40 μm. Immunoblot shows the knockdown efficiency of SUN2. The bar graph shows percent nuclear deformation in siLUC and siSUN2 cells. $N = 576$ cells in siLUC and 623 cells in siSUN2. **b** Confocal images of DAPI staining showing nuclear deformation in control (siLUC), SUN1 depleted (siSUN1), PREP1 depleted (siPREP1) or PREP1 and SUN1 depleted (siPREP1 + siSUN1) U2OS cells, as indicated. Scale bar is 20 μm. White lines in the blot image indicate where an in-between lane was spliced out. The bar graph shows percent nuclear deformation in siLUC, siPREP1 and siPREP1 + siSUN1 conditions. $N = 737$ cells for siLUC, 670 cells for siPREP1, 551 cells for siSUN1 and 771 cells for siPREP1 + siSUN1. **c** Representative confocal images of control (siLUC) and PREP1 depleted (siPREP1) U2OS cells showing F-actin (Phalloidin, green) and nucleus (DAPI, blue). Scale bar is 20 μm. **d** Maximum projection of apical (upper panel showing perinuclear actin stress fibers) or basal (lower panel showing ventral actin stress fibers) confocal stacks of control (siLUC) and PREP1 depleted (siPREP1) HeLa and U2OS cells. Images show F-actin (Phalloidin in green) and nucleus (DAPI in red). Scale bar is 20 μm. **e** Maximum projection of confocal images of F-actin (Phalloidin, green) and nucleus (DAPI, red) in control (siLUC) and PREP1 depleted (siPREP) U2OS cells grown in fibronectin-coated micropatterns. Right panel shows orthogonal view of the same cells. Scale bar is 20 μm. Bar graph shows the percentage of cells with perinuclear actin cap in siLUC and siPREP1 conditions. $N = 33$ cells. $p = 0.0156$. **f** Maximum projection of confocal z stacks showing nuclear envelope (LAMIN A/C, red) and F-actin (Phalloidin, green) of blebbistatin treated or vehicle (+DMSO) of control (siLUC) and PREP1 depleted (siPREP1) U2OS cells. Scale bar is 40 μm. The bar graph shows quantification of severe nuclear deformation in siLUC and siPREP1 U2OS cells upon treatment with 40 μM Blebbistatin for 20 min. Error bars represent mean ± SE of three independent experiments. $N = 898$ (siLUC DMSO), 878 (siLUC Blebbistatin), 1067 (siPREP1 DMSO) and 890 cells (siPREP1 Blebbistatin). $p = 0.0473$.

transfected cells with pEGFP-KASH2 construct[38] which unlike pEGFP-KASH2ext, is expected to disrupt the interaction between nesprins and nuclear membrane, and analyzed the actin fibers and focal adhesion size. As expected, in KASH2 expressing cells, NESPRIN2 was displaced from the nuclear envelope indicating a disruption of the LINC complex, while KASH2ext expressing cells displayed the typical nuclear envelope localization of NESPRIN2 (Supplementary Fig. 4d). However, we did not detect any change in actomyosin bundles (Supplementary Fig. 4e) or vinculin foci in KASH2 expressing U2OS cells (Fig. 4d). Additionally, downregulation of SUN2 also did not cause visible perturbation of actin bundles (Supplementary Fig. 4f) or vinculin foci (Fig. 4e).

The SUN2/SUN1 status of the cell can affect the actin dynamics through the SRF/Mkl1 transcription factor[39–42]. However, in PREP1 depleted cells we did not detect any significant change in transcript levels of *VINCULIN* or *SM22*, two SRF/Mkl1 target genes (Supplementary Fig. 4g). All these data show that a general perturbation of the LINC complex or SUN2 downregulation is not sufficient to induce the cytoskeletal changes induced by PREP1 downregulation.

Focal adhesion molecules like Src, FAK and p130CAS act as mechanosensors and the phosphorylation of these proteins signals the functionality of focal adhesions[43–46]. Indeed, increased phosphorylation of Src and p130CAS were detected in HeLa and U2OS cells upon PREP1 depletion (Fig. 4f and Supplementary Fig. 4h), indicating the higher tension experienced by these cells in the absence of any extracellular cues. All these data prompted us to ask if PREP1 depleted cells can sense substrate rigidity correctly.

**PREP1 acts upstream of YAP-TAZ mechanotransduction.** Mechanosensing of substrate rigidity is the initial and essential part of mechanosignaling and triggers nuclear translocation of transducers such as YAP-TAZ[47]. YAP-TAZ is translocated to the nucleus in response to increases in extracellular matrix rigidity, focal adhesion complexes and actomyosin contractility[47,48]. To probe if bigger focal adhesions and increased actomyosin contractility translate into increased mechanotransduction, we looked at YAP-TAZ localization. Contrary to our expectations, in U2OS cells grown on glass coverslips, PREP1 depletion caused cytoplasmic retention of YAP-TAZ compared to control cells as shown by immuno-fluorescence analysis (Fig. 5a). To compare and contrast siPREP1 cells with single depletions of SUN1 and SUN2, we assessed the YAP translocation in SUN1 or SUN2 depleted U2OS cells grown on glass coverslips. Downregulation of SUN1/SUN2 in various cell types has been shown to affect actin

cytoskeleton in a cell type dependent manner[49,50]. In U2OS cells, depletion of SUN1 did have minimal effect on YAP nuclear translocation compared to SUN2 depleted U2OS cells where the effect was more substantial (Supplementary Fig. 5a, b). However, this result is opposed to PREP1 depleted cells where increased FA signaling and actin fibers did not favor nuclear translocation of YAP. These data again showed that cellular phenotypes of PREP1 is more complex and nuanced than what happens in single depletions of SUN1/2.

Further, to understand if PREP1 depleted cells can sense the change in substrate rigidity, we grew PREP1 depleted U2OS cells in soft (0.5kPa) vs rigid (normal cell culture plates) substrates and assessed the localization of YAP by immunofluorescence analysis (Fig. 5b). Nuclear localization of YAP was reduced in cells grown in soft matrix as expected[48]. However, PREP1 depleted cells demonstrated more cytosolic YAP localization in both conditions (Fig. 5c) indicating faulty mechanotransduction. We also probed the phosphorylation of serine 127 residue in YAP which has been shown to induce 14-3-3 protein binding and hence its cytoplasmic localization[51]. Immunoblotting detected increased phospho-YAP-TAZ in PREP1 depleted cells in both soft and rigid substrates indicating cytoplasmic YAP localization in both conditions (Fig. 5c). Thus, despite an increase in cellular traction forces or actomyosin contractility, YAP-TAZ transduction was impaired in PREP1 depleted cells.

The fact that PREP1 transcriptionally can regulate SUN1, SUN2 and LAP2 in a variety of cells of mouse and human origin prompted us to wonder whether this was a (conserved) mechanism employed by the cell to regulate nuclear integrity. Therefore, we asked whether a cell experiencing an insult to its nuclear membrane integrity, would increase PREP1 expression. To address this, we downregulated SUN1 or SUN2 in U2OS and HeLa cells and assessed PREP1 expression. Intriguingly, we detected increased PREP1 in both SUN1 and SUN2 depleted cells (Fig. 5d) indicative of a feedback mechanism. qPCR experiments detected a slight but consistent increase in *PREP1* mRNA levels, indicating possible transcriptional regulation of PREP1 upon SUN1 or SUN2 downregulation (Fig. 5e). Interestingly, SUN2 was downregulated in SUN1 depleted U2OS cells as well (Fig. 5d), raising the question whether the feedback loop is connected to the transcriptional output of SUN2 rather than a change in nuclear envelope stoichiometry. To address this point, we downregulated LAMIN B1 whose transcript and protein expression was not affected by PREP1 in U2OS cells, using RNA interference. LAMIN B1 depletion did not change SUN2 protein levels (Fig. 5f). However, PREP1 expression was still high in these cells indicating that PREP1

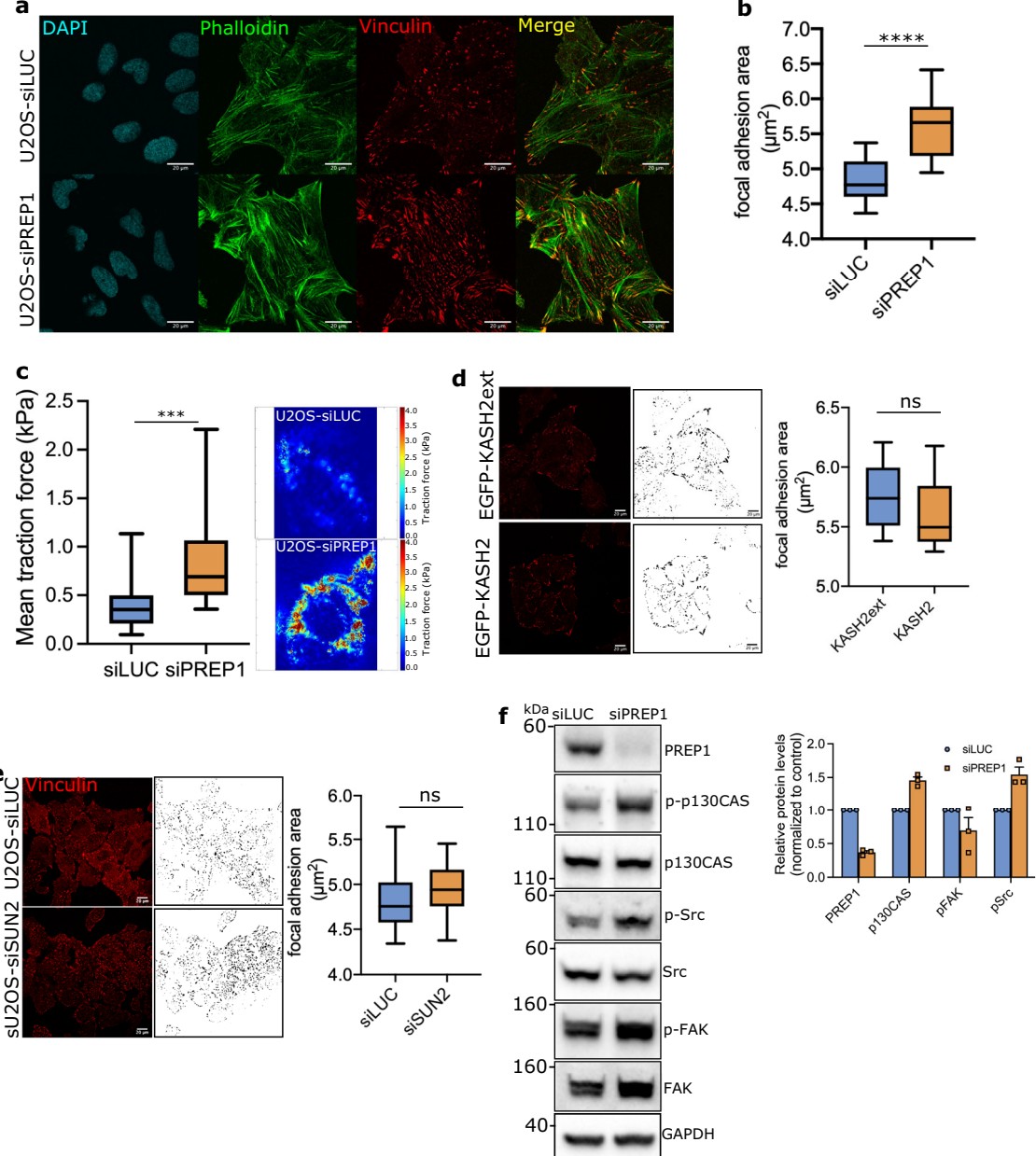

**Fig. 4 PREP1 depletion affects the focal adhesion pathway. a** Representative confocal images showing adhesion foci (vinculin, red) and actin stress fibers (Phalloidin, green) in control (siLUC) and PREP1 depleted (siPREP1) U2OS cells. Nuclei are shown in blue (DAPI). Scale bar is 20 μm. **b** Box and whisker plot show quantification of focal adhesion area in control (siLUC) and PREP1 depleted (siPREP1) U2OS cells. Graph is derived from four independent experiments. $N = 572$ cells in siLUC and 611 cells in siPREP1. **c** Representative traction force heat-map images of control (siLUC) and PREP1 depleted (siPREP1) U2OS cells. The box plot shows mean traction force in kPa experienced by siLUC or siPREP1 U2OS cells. The data is derived from three independent experiments. $N = 25$ cells each in siLUC and siPREP1 cells. **d** Confocal images of vinculin foci and its corresponding binary image in U2OS cells expressing EGFP-KASH2ext or EGFP-KASH2. Box plot depicts the quantification of focal adhesion area in EGFP-KASH2ext or EGFP-KASH expressing U2OS cells. Data is derived from two independent experiments. $N = 101$ cells in EGFP-KASH2 and 120 cells in EGFP-KASH2ext. **e** Confocal images of vinculin foci and its corresponding binary image in control (siLUC) and SUN2 depleted (siSUN2) U2OS cells. Box plot shows focal adhesion area in the same cells. Data is derived from three independent experiments. $N = 433$ cells in siLUC and 628 cells in siSUN2. **f** Representative immunoblot showing the phosphorylation status of p130CAS, Src and FAK proteins in whole cell lysates of control (siLUC) and PREP1 depleted (siPREP1) U2OS cells. Corresponding total protein levels and GAPDH serve as loading control. Blot is representative of three independent experiments. Bar graph shows the relative protein expression in PREP1 depleted cells normalized to control cells across experiments. Error bars denote mean ± SE.

expression is not linked to SUN2 but to the change in nuclear envelope stoichiometry (Fig. 5f). These data suggest the presence of a transcriptional feedback mechanism of PREP1 upon challenges to nuclear integrity. Further, we propose that nucleo-cytoskeletal coupling by PREP1 is necessary for the correct mechanotransduction of YAP-TAZ.

## Discussion

Transduction of extracellular cues to signaling pathways inside a cell by mechanosignaling is an important aspect of development and differentiation[7,8,52]. Our data reveal a transcriptional regulation of mechanosignaling at the nucleo-cytoskeletal coupling. We show that transcription factor PREP1, a cofactor for Hox

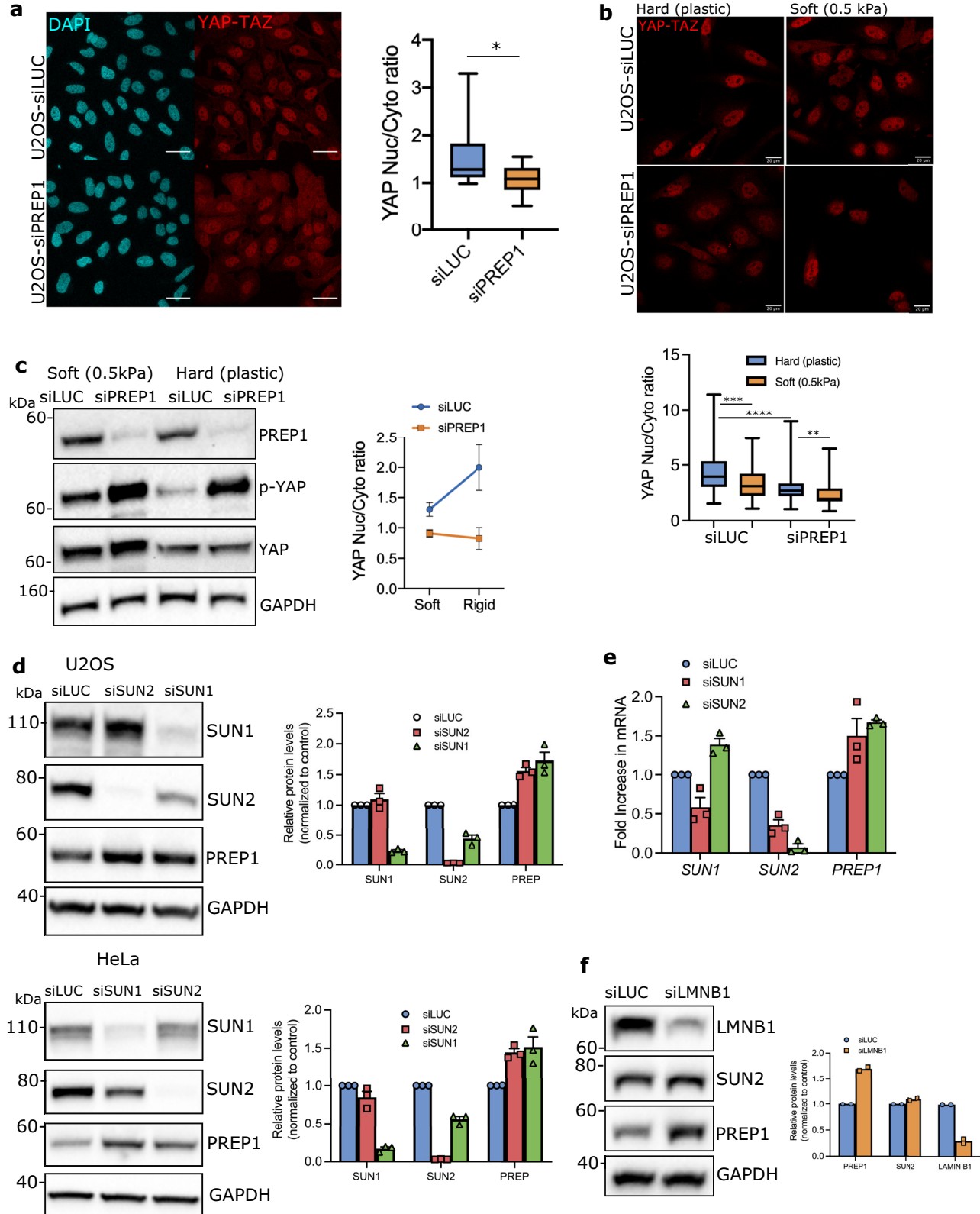

proteins in determining axial patterning, is required for the transcriptional regulation of Nuclear Envelope proteins SUN1, SUN2 and LAP2, thereby influencing nuclear mechanics. Loss of PREP1 renders the nuclei softer because of the perturbation in the level of these proteins, and reduces the localization of Lamin A/C to the nuclear lamina. Seminal studies have shown the importance of nuclear lamina in receiving and responding to the extracellular mechanical cues driving lineage specification. Lamin A/C expression increases with tissue stiffness and contributes to lineage commitment during matrix directed differentiation of MSCs[53]. In MSCs, a mechanical equilibrium between Lamin A/C and myosin IIA is maintained through a mechanosensitive feedback loop[54]. In such a system, cells grown on a stiff matrix display high Lamin A/C content, increased stress fibers and

**Fig. 5 PREP1 affects YAP-TAZ mechanotransduction. a** Immunofluorescence images showing YAP-TAZ localization in control (siLUC) and PREP1 depleted (siPREP1) U2OS cells. Scale bar is 40 μm. Box plot shows the ratio of nuclear to cytoplasmic fluorescence intensity of YAP-TAZ in two independent experiments. $N = 687$ cells for siLUC and 567 cells for siPREP1. **b** Immunofluorescence images showing YAP-TAZ localization in control (siLUC) and PREP1 depleted (siPREP1) cells grown on soft (0.5kPa) or hard substratum (normal cell culture dishes). Scale bar is 20 μ. Box plot below shows the nuclear to cytoplasmic ratio of YAP-TAZ. $N = 68$ (siLUC hard), 81 (siLUC soft), 88 (siPREP1 hard) and 48 cells (siPREP1 soft). $p = 0.031$ (**), $p = 0.0003$ (***) and $p < 0.0001$ (****). **c** A representative immunoblot showing s-127 phosphorylation status of YAP-TAZ in control (siLUC) and PREP1 depleted (siPREP1) U2OS cells grown in soft (0.5kPa) vs hard (plastic) substrate. Total YAP and GAPDH act as loading controls. The data is representative of three independent experiments. Graph below shows the ratio of total YAP to phosphorylated YAP. Error values represent mean ± SE. **d** A representative immunoblot showing total PREP1 levels in control (siLUC), SUN1 depleted (siSUN1) or SUN2 depleted (siSUN2) U2OS or HeLa cells. GAPDH serves as loading control. Data is representative of three independent experiments. Corresponding bar graph shows densitometric analysis of band intensity from three independent experiments. Error bars represent mean ± SEM. **e** Bar graph shows *PREP1* mRNA transcript levels in siLUC, siSUN1 or siSUN2 U2OS cells. Graph is derived from three independent experiments. Error values represent mean ± SE. **f** A representative immunoblot showing total PREP1 levels in control (siLUC) and LAMIN B1 depleted (siLMNB1) U2OS cells. Data is representative of two independent experiments. Bar graph shows the densitometric analysis from two experiments.

increased nuclear translocation of YAP[53,55]. Our experiments show that PREP1 depletion uncouples the mechanosensitive feedback loop between the nuclear lamina and actomyosin structures and inhibits nuclear translocation of YAP in stiff substrates. Such a profound effect on mechanosignaling is bound to have far reaching consequences in embryogenesis and differentiation. Indeed, the Prep1 knockout mutation is early embryonic lethal around day 6 post fertilization[2]. Hypomorphic mice with a less penetrant phenotype (Prep1i/i cells express 3–10% Prep1 protein) survive embryonic lethality but develop tumors later in life[56]. Bone marrow derived MSCs from these mice have a propensity to differentiate into adipocytes even in the absence of adipogenic stimuli[5]. Further, Prep1 hypomorphic mice are characterized by aberrant fat deposition in the body and lower bone density because of the inherent tendency of PREP1 deficient progenitor cells towards the adipocytic rather than osteocytic lineage[6]. All these data fit well with our observation that PREP1 depletion indeed inhibits the propagation of matrix derived, cytoskeleton mediated mechanical cues. A disruption of the nucleo-cytoskeleton link might also impair other nuclear shuttling proteins such as β-catenin, involved in mechanosignaling. It is interesting to note that the Wnt/β-catenin axis is impaired in ES cells of *Prep1* KO mice[4]. Thus, PREP1 appears to have an important role in signal transduction during mechanosignaling.

PREP1 depletion causes similar changes in the expression pattern of nuclear envelope proteins in both HeLa and U2OS cells, and in both cell lines this is reflected in a decreased stiffness of the nucleus. However, nuclear deformation, i.e. the phenotypical presentation of this defect, varies from HeLa to U2OS since actomyosin induced stress varies in these cells (Fig. 6). Our data show that nuclear morphology in PREP1 depleted U2OS cells is partly regulated by the perinuclear actin cap which, indeed, has been shown to have a major role in interphase nuclear morphology[30]. Moreover, the fact that PREP1 depletion triggers perinuclear actin cap formation in U2OS cells is interesting since cancer cells, like U2OS, are devoid of actin caps[57]. How and why this change happens and its implications in tumor biology and cell migration need to be addressed in future studies.

While we see clear changes in the nuclear LAMIN A/C intensity by immunofluorescence analysis, immunoblotting analysis for protein and qPCR analysis for mRNA, show that LAMIN A/C protein levels are not changed. Notwithstanding the fact that quantitative use of IF analysis is dependent on the image acquisition parameters and is not as efficient as immunoblotting, we speculate that the difference in epitope masking/antibody binding efficiency in a denatured vs native state in immunoblotting and immunostaining respectively might be the reason for this discrepancy. LAMIN A/C epitope accessibility has been shown to be dependent on LAMIN A/C multimerization, functional LINC complexes and cytoskeletal forces[58]. Our data shows that PREP1 depletion leads to changes in the components of LINC complex and actin cytoskeleton and hence it is possible that epitope accessibility is different in these cells leading to differential outcome in immunofluorescence analysis.

The LINC complex has been shown to be essential in perinuclear actin cap formation[30,57]. However, it is possible that the increased SUN1 is able to compensate for the loss of SUN2 and to form fully functional actin caps. SUN2 has been proposed as a key intermediate in decoupling nucleus from the cytoplasm during cyclic tensile strain in primary mesenchymal stem cells[59]. However, uncoupling nucleus and cytoskeleton by introducing the EGFP-KASH2 construct or by selective SUN2 knockdown also doesn't lead to a general increase in focal adhesions or actin cytoskeleton in our experimental system, unlike what is seen in PREP1 depletion, clearly suggesting that the transcriptional regulation by PREP1 is specific and more nuanced. It is possible that the buildup of actomyosin bundles and focal adhesions is a compensatory effect due to the changes in the nuclear envelope proteins.

Nuclear accumulation of YAP-TAZ is regulated by rigidity and topology of the ECM substrate, the status of Src/FAK signaling and actomyosin contractility[48,60–62]. However, in PREP1 depleted cells, irrespective of increased actomyosin bundles and highly efficient focal adhesion signaling, YAP is inefficiently translocated to the nucleus, demonstrating the impaired mechanotransduction in these cells.

Our study shows the existence of a cellular transcriptional control over the mechanotransducer YAP-TAZ irrespective of the rigidity of the extra-cellular matrix. While we cannot rule out the influence of LATS mediated serine 127 phosphorylation of YAP as the first event which leads to its cytoplasmic retention, it is interesting to note that a recent study brought to light the importance of force application to nucleus alone for efficient YAP translocation[63]. It is possible that a more fragile nucleus in PREP1 depleted cells is also not responsive to the force-induced nuclear pore stretching leading to cytoplasmic retention of YAP.

Our data also bring to light the presence of a negative feedback loop mechanism between PREP1 and the nuclear envelope protein stoichiometry: upon SUN1 or SUN2 depletion, cells upregulate PREP1 protein. While the data is still preliminary, a similar upregulation of PREP1 upon LAMIN B1 depletion indicates that the feedback loop is linked to the integrity of nuclear lamina/nuclear envelope and not to the transcriptional reduction of a specific protein. The mechanism behind the feedback loop and YAP-TAZ cytoplasmic retention requires a detailed analysis. We envisage the existence of a cellular transcriptional mechanism which can intervene to actively regulate nuclear membrane

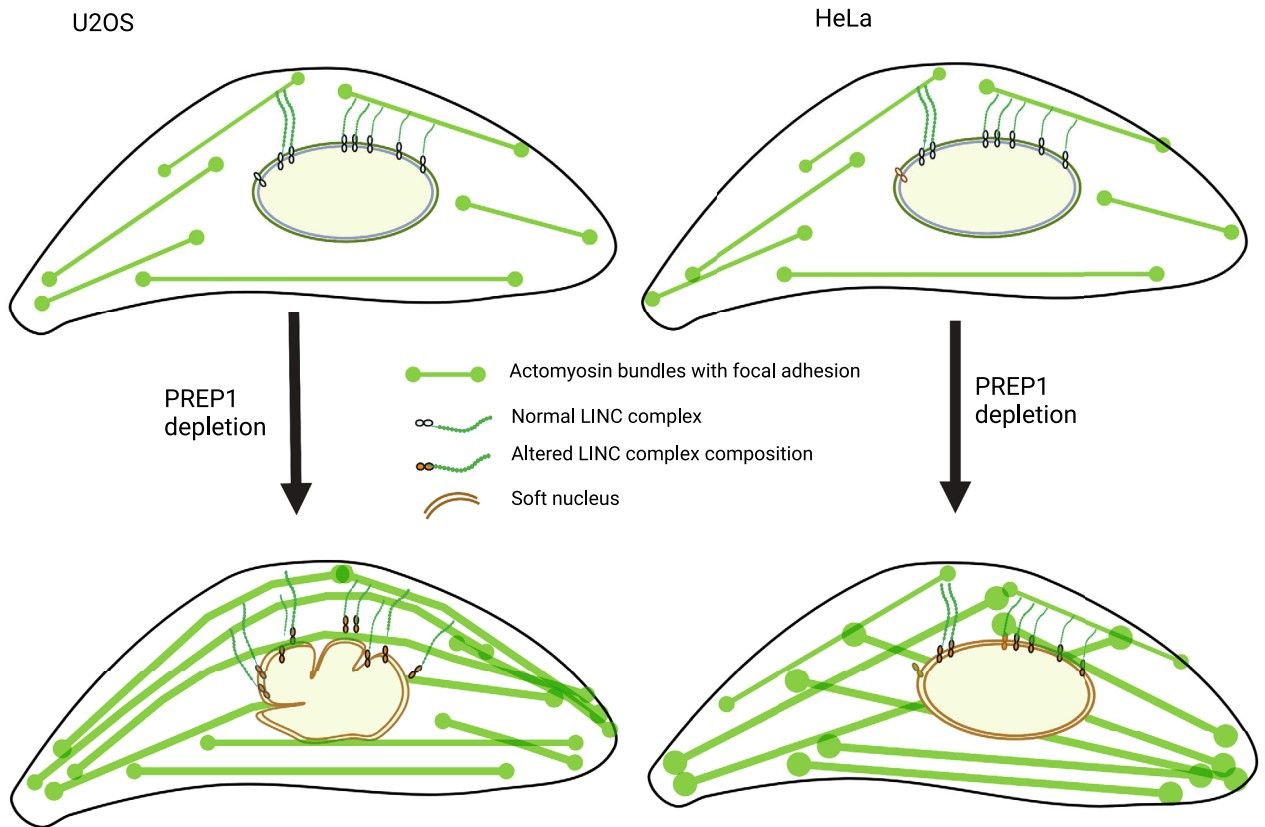

**Fig. 6 Schematic representation of the effects of PREP1 depletion in U2OS vs HeLa cells.** Schematic view (created with BioRender.com) showing the differential response of U2OS and HeLa cells to PREP1 depletion.

homeostasis and signal transduction by the active engagement/disengagement from the extracellular matrix.

## Methods

**Cell culture**. HeLa cells (ATCC) were cultured in MEM (Lonza) reconstituted with 10% heat-inactivated FBS (Sigma-Aldrich), 2mM L-Glutamine, 1 mM Sodium Pyruvate and 0.1 mM Non-essential amino acids (Bio West). U2OS (ATCC) were cultured in DMEM (Lonza) reconstituted with 10% heat-inactivated FBS (Sigma-Aldrich) and 2mM L-Glutamine. Cell lines were tested for mycoplasma contamination.

Cardiac mesenchymal stromal cells used in ChIP-seq experiments were obtained from the right ventricular free wall endomyocardial biopsies of healthy controls (project CCM1072, approved by Centro Cardiologico Monzino ethic committee and amended on the 02/12/2020) and propagated in TMES medium as reported previously[57].

Cytosoft cell culture plates (Advanced Biomatrix, CA) with PDMS substrates of 0.5 kPa were used for assessing YAP-TAZ translocation in response to substrate stiffness.

**Transfections and RNAi**. Lipofectamine RNAiMAX (Invitrogen) was used to transfect 10 nM dsRNA following manufacturers protocol. 48 h post transfection, cells were used for assays. To down regulate PREP1, a cocktail of two siRNAs were used (siRNA 607 = GAUUUCUGCAGUCGAUACA; siRNA 900 = CUCCCAGCUUCAGUUACAG)[11]. To target LMNB1 10 nM siRNA duplexes of abx903005 (Abbexa Ltd LMNB1: GCAGACUUACCAUG CCAAA) was used. To target SUN1, a Mission siRNA (Sigma-Aldrich SASI_Hs01_00032809) with the sequence CAGCUAAAGUCAGAGCUGU was used. To target SUN2 a Mission siRNA (Sigma-Aldrich SASI_Hs01_00176980) with the sequence CUAUUCAGACGUUUUCACUU was used. An siRNA targeting firefly luciferase (CAUCACGUACGCGGAAUAC) was used as control in all experiments.

**Retroviral infections and development of stable cell lines**. Phoenix amphotropic cell line was used to generate virus particles. pBABE puro-EV or pBABE puro-PREP1 plasmids were transfected using standard Calcium phosphate

mediated transfection method for virus production. Cell culture supernatant containing virus particles were concentrated using the PEG-it virus precipitation solution (System Biosciences, CA). Standard polybrene enhanced retroviral infection was used for the stable integration of the plasmids in A549 cell line. 24 h after infection, cells with stable viral integration was enriched by puromycin antibiotic selection.

**Antibodies**. Antibodies used were from the following sources.

IF: LAMIN B1 (Goat, 1:400, Santa Cruz Biotechnology), LAMIN A/C (Mouse, 1:1000, Santa Cruz Biotechnology), SUN1 (Rabbit, 1:200, Abcam), SUN2 (Rabbit, 1:200, Abcam), LAP2α (Rabbit, 1:400, Abcam), LAP2β (Mouse, 1:200, Sigma), LBR (Rabbit, 1:400, Abcam), EMERIN (Mouse, 1:500, Leica), PREP1 (Mouse, 1:200, Santa Cruz Biotechnology), vinculin (Mouse, 1:1000, Sigma-Aldrich), YAP-TAZ (Mouse, 1:200, Sigma-Aldrich), Nesprin1 (Mouse, 1:200, ThermoFisher Scientific), Nesprin-2 (Mouse, 1:200, Immuquest).

WB: LAMIN B1 (Rabbit, 1;10,000, Abcam), LAMIN A/C (Mouse, 1:10,000, Santa Cruz Biotechnology), SUN1 (Rabbit, 1:10,000, Abcam), SUN2 (Mouse, 1:10,000, Abcam), LAP2α (Rabbit, 1:10,000, Abcam), LAP2β (Mouse, 1:5000, Sigma), LBR (Rabbit, 1:10,000, Abcam), EMERIN (Mouse, 1:10,000, Leica), PREP1 (Mouse, 1:3000, Santa Cruz Biotechnology), Nesprin1 (Mouse, 1:2500, ThermoFisher Scientific), p-FAK (Y397) (Mouse, 1:5000, Cell Signaling Technologies), tot-FAK (Mouse, 1:1000, Cell Signaling Technologies), p-p130CAS (Y410) (Rabbit, 1:2500, Cell Signaling Technologies), tot-p130CAS (Rabbit, 1:1000, Cell Signaling Technologies), p-Src (Y416) (Rabbit, 1:1000, Cell Signaling Technologies), tot-Src (Mouse, 1:1000, Merck Millipore), GAPDH (Mouse, 1:10,000, Santa Cruz Biotechnology), YAP-TAZ (Mouse, 1: 1000, Santa Cruz Biotechnology), p-YAP (1:1000, Cell Signaling Technologies).

**Immunofluorescence (IF)**. For Immunofluorescence analysis of nuclear envelope proteins, cells were grown on coverslips, transfected with siRNA for 48 h, fixed with 4% paraformaldehyde in 1x PBS (Sigma-Aldrich) for 10 min, permeabilized with 0.1% Triton X-100 in PBS (v/v; Euroclone) and blocked in 5% Donkey serum for 1 h at RT. Primary and secondary antibodies were incubated in blocking solution for 1 h/1.5 h at RT. Nuclei were visualized by DAPI staining. Images were acquired using a 63X lens on a Leica confocal microscope SP2 or SP5. Acquired images were analyzed by using the methods described below.

## Image analysis

*Nuclear Intensity quantification of Lamins and other nuclear envelope proteins.* In order to quantify the protein intensity of lamins and other nuclear envelope proteins, a custom FIJI plugin was developed. The nuclei were identified with DAPI channel, filtered with a Gaussian filter, applying ImageJ Default threshold method followed by watershed to separate the nuclei. Nuclear lamina was identified with either Lamin A or Lamin B channel as the overlap of the regions found using ImageJ Moments threshold method on the raw image and a filtered image (with a gray morphology circle filter and a Gaussian filter). Lamina regions were then separated with watershed and each lamina assigned to a nucleus if there is overlap between the nucleus and the lamina. The plugin was used to measure the mean intensity of proteins at the lamina and also the mean intensity of proteins in the nucleus.

## Focal adhesion analysis and quantification.

In order to visualize focal adhesions, cells were stained with anti-vinculin antibody. Using the custom-built ImageJ macro, the best z-plane was selected, the signal was segmented using Moments (value 25) and the size of the particles was measured. For each experimental condition, an average area of focal adhesions was calculated.

## Estimation of YAP-TAZ nuclear/Cytoplasmic ratio.

Nuclear/cytosolic ratio of YAP was assessed by measuring the intensity of an equal area region inside the nucleus and immediately outside the nucleus in ImageJ and calculating the ratio as previously published[63]. DAPI staining was used for delimiting the nucleus.

## Quantification of nuclear shape description parameters.

For quantification of nuclear shape parameters, U2OS cells transfected with control and PREP1 siRNAs were manually seeded on black clear-bottom CellCarrier 384-Ultra microplates-Perkin-Elmer (2500-3000-3500 cells per well) 24 h after transfection. Twenty-four hours later, cells were fixed and processed immediately for immunofluorescence. Briefly, cells were fixed with 4% paraformaldehyde for 15 min, permeabilized with 0.2% Triton X-100 in phosphate-buffered saline (PBS) for 15 min, followed by 40 min blocking in PBS-1% BSA-0.1% Tween-20. Cells were then incubated with a mouse antibody against Lamin A/C (Santa Cruz Biotechnology) diluted in blocking solution for 1 h. Cells were further washed with PBS and incubated for 1 h with a secondary antibody conjugated to Alexa Fluor-488 (Life Technologies) and stained with DAPI (Life Technologies). All cell staining procedures were automated on Wellwash Versa (Thermo Scientific) and Multidrop Combi (Thermo Scientific).

Using an Operetta high-throughput spinning-disk confocal microscopy system (Perkin Elmer), cells were imaged with a 63× water immersion objective. Captured images were analyzed using custom-developed Acapella Software (Perkin Elmer) image analysis algorithms. An internal developed image pipeline for the nuclear perturbation analysis was used to enumerate the following parameters: nuclei-roundness, axial length ratio, width to length ratio, Lamin A/C invaginations (lamin fragmentation index), and the percentage of nuclei with nuclear membrane invaginations.

## Western blotting.

Western blot analysis was performed following standard procedures. Briefly, whole-cell lysates were made in Laemmli buffer and the samples were resolved using Invitrogen Bolt 4–12% Bis-Tris gels (ThermoFisher Scientific). Proteins were transferred to Amersham Protran nitrocellulose membrane using standard protocols and blocked in 5% milk in TBS-Tween (0.2%). Primary antibodies were incubated at 4 °C overnight, followed by washes in TBS-T. The corresponding secondary antibodies conjugated to HRP (Horshradish peroxidase) were incubated for 1 h at room temperature. Blots were developed using Supersignal West Dura/Femto/Pico substrates (ThermoFisher Scientific) using Chemidoc XRS + (BioRad).

## RNA extraction, cDNA synthesis, and qPCR.

RNA was extracted using Qiagen RNeasy kit (Qiagen) and cDNA synthesis was done using Superscript III (Invitrogen) according to manufacturer's protocols. qPCR was done using Roche LightCycler 480 SYBR Green I master mix in a LightCycler 96 (Roche) with the following primers.

| Gene of interest | qPCR primers sequence |
|---|---|
| PREP1 | Forward:5'CTGCAGCAGGGAAACGTAG |
|  | Reverse: ACCGTGACAGGCTGATACACT |
| SUN1 | Forward: ATCCCGCTGTGGTACTTCTC |
|  | Reverse:AATGCCCAGCAGTTACCG |
| SUN2 | Forward: GGCGCGGTGACTTAGA |
|  | Reverse:GTCCTGCTGAAGGAGGTGAC |
| LMNB1 | Forward: TGGGAAATTTATCCGCTTGA |
|  | Reverse: TGACTGATGTGTCTCCAATTTTTT |
| LMNA | Forward: GCCATCGACAGCCTCTCT |
|  | Reverse:GTCCTCCAGGTCTCGAAGC |
| LBR | Forward: TGGCAGTGAGAACCTTTGAA |
|  | Reverse:CAGGCCAAACATGATGAGAA |
| LAP2 | Forward: ACCATTGACAAGAGCTGAAGTG |
|  | Reverse: GAACATTTCCTTAAGAATATCCCTTTC |
| SYNE2 | Forward: CAGCCTCCTGCAACATCC |
|  | Reverse: AGAGGAAGGAGCGCTGTG |

## Chromatin immunoprecipitation.

Chromatin immunoprecipitations (IP) were performed using standard methods with anti-PREP1 N15 antibody (sc-6245, Santa Cruz Biotechnology, Santa Cruz, USA). Cells were cross-linked in complete medium (10% FBS) containing 1% formaldehyde for 10 min, and glycine was added to stop the reaction (125 mM final concentration). Fixed cells were washed three times (5 min each) in cold PBS and lysed in LB1 buffer (LB2 buffer containing 0.5% NP-40 and 0.25% Triton X-100). Nuclei were washed in LB2 buffer (10 mM Tris-HCl pH 8 and 200 mM NaCl) to remove detergents and resuspended in LB3 buffer (LB2 buffer containing 0.1% Na-deoxycholate and 0.5% N-lauroylsarcosine). Chromatin was sonicated in Covaris ultrasonicator to obtain fragments ranging 150–250 bps. Sonicated chromatin was incubated with antibody-bound protein G-conjugated magnetic beads (Invitrogen, Carlsbad, USA). For each IP we used 5 µg antibody. IP with rabbit IgG was performed as negative control. After overnight IP at 4 °C the bound complexes were washed twice in WB1 (50 mM Hepes-KOH pH 7.5, 140 mM NaCl, 1 mM EDTA, 1% Triton X-100, 0.1% Na-doexycholate), twice in WB2 (50 mM Hepes-KOH pH 7.5, 500 mM NaCl, 1 mM EDTA, 1% Triton X-100, 0.1% Na-deoxycholate) and twice in LiCl WB (10 mM Tris-Cl pH 8.0, 250 mM LiCl, 0.5% NP-40, 0.5% Na-deoxycholate, 1 mM EDTA). Immunoprecipitated complexes were eluted from the beads by incubation for 30 min in EB (2% SDS in TE) at 37 °C. The eluted material was reverse cross-linked at 65 °C overnight and incubated for 1 h at 55 °C with proteinase K. DNA was purified with a PCR purification kit (Qiagen, Netherlands). About 1/10 of the immunoprecipitated DNA was used for qPCR. The following primers, specific to the corresponding promoter regions, containing putative PREP1 binding sites, were used.

SUN2: Forward GACTCTGTATGTGTGCGCCT; Reverse TTCAAACCGGCCAATGGGT

SUN1: Forward ACACTTGGGCTGTTTGCAC; Reverse TGACCTGGCAACTCCATTC

LAP2 (TMPO): Forward TCGGATGATGGACAGCAAGTG; Reverse GCTCCTTTTTCATCCCTGGTG

The qPCR reaction was performed on the Roche Lightcycler 480® in three independent biological replicates. The enrichment factor was calculated after the measurement of the amount of the immunoprecipitated material.

## AFM indentation method.

AFM indentation was carried out using JPK Nano-Wizard3 mounted on a Olympus inverted microscope. A modified AFM tip (NovaScan, USA) attached with 10 µm diameter bead was used to indent the center of the cell. The spring constant of the AFM tip cantilever is ~0.03 N/m. AFM indentation loading rate is 0.5 Hz with a ramp size of 3 µm. AFM Indentation force was set at a threshold of 2nN. The data points below 0.5 µm indentation depth were used to calculate Young's modulus to ensure small deformation and minimize substrate contributions. The Hertz model is shown below:

$$F = \frac{4}{3}\frac{E}{(1 - \nu^2)}\sqrt{R\delta^3}$$

where $F$ is the indentation force, $E$ is the Young's modulus to be determined, $\nu$ is the Poisson's ratio, $R$ is the radius of the spherical bead, and $\delta$ is indentation depth. The cell was assumed incompressible and a Poisson's ratio of 0.5 was used.

## Micro-patterning.

Micropatterns of fibronectin-coated lines (10 µm of width) were fabricated using photolithography as previously described[64]. Briefly, the glass surface of the coverslip was activated with plasma cleaner (Harrick Plasma, 1 min HIGH setting) and then coated for 1 h at RT with PLL-g-PEG (Surface Solutions GmbH, 0.1 mg/mL in 10 mM HEPES, pH 7.4). After washing with 1X PBS and deionized water, the surface was illuminated for 7 min with deep UV light (UVO Cleaner, Jelight) through a quartz photomask (Delta mask B.V.). Micro-patterned coverslips were then incubated 1 h at 37 °C with fibronectin solution (10 µg/mL in 1x PBS, Sigma-Aldrich). Extra fibronectin was removed washing coverslips with 1x PBS. 32 h post transfection, cells were detached using trypsin-EDTA (1x in PBS, Euroclone) and left 16 h to attach on micro-patterned lines (10,000 cells/coverslip).

## Traction force microscopy.

TFM analyses were performed as previously reported[65,66]. Briefly, U2OS and HeLa cells were seeded and incubated for 16 h on fibronectin-coated silicone samples containing highly regular arrays of Red fluorescent quantum dots (QDs) yielding an elastic modulus of 5, 16 kPa. Images were acquired using a 40x objective (Leica Germany) with a Leica TCS SP5 confocal microscopy (Leica Instruments) to reconstruct the traction field. Single cells were cropped and QDs displacement from resting positions was analyzed through the Cellogram software[67] using the known properties of the material.

## Statistics and reproducibility.

Experiments were repeated minimum three times unless stated otherwise and is indicated in the figure legends. P-values were determined using unpaired Students T-test using Graphpad Prism.

**Reporting summary**. Further information on research design is available in the Nature Research Reporting Summary linked to this article.

## Data availability

Chip-seq data on human cardiac mesenchymal stem cells generated during this study have been deposited at GEO and is retrievable with the accession code GSE160286. Chip-seq data on Hela cells (accession code GSE101776), mouse embryos (accession code GSE39609), and mouse embryonic stem cells (accession code GSE63282) are available at GEO database. The source data for all the graphs prepared for the manuscript is available as Supplementary Data 1. Uncropped blots used for preparing figures in the manuscript are available as Supplementary Fig. 6. All other data are available from the corresponding authors upon reasonable request.

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

## Acknowledgements

D. Pu. and F. B. are very grateful to GV Shivashankar for his interest in the work and helpful discussions and suggestions. EGFP-KASH2 and EGFP-KASH2ext constructs were a kind gift from Didier Hodzic. We thank Emanule Martini at IFOM imaging facility, Adrian Andronache and Moni Cera of ETP, IFOM for developing the pipelines for image analysis and for help with experiments. We thank Stefano Piccolo for help with YAP-TAZ antibodies and protocols. We are also grateful to Stefano Casola, Kristina Havas, Colin Stewart and Brian Burk for useful discussions. This work was supported by the AIRC (Italian Association for Cancer Research) grant (2015-16759), and Cariplo grant (Grant 2018-0520). D. Pe. was supported by a grant from Russian Foundation of Basic Research (19-29-04-112). F. A .P was supported by a fellowship from the Italian Association for Cancer Research (23966).

## Author contributions

D. Pu. conceptualized and supervised the project, designed and performed the experiments, analyzed the data, and wrote the manuscript. L.F.B. performed the experiments and analyzed the data., D. Pe. performed Chip and ChipSeq experiments. Q.L. performed AFM experiments. A.P. contributed to micro-patterning experiments. A.P., F.M.P., and F.A.P. performed traction force experiments. J.V. and R.C. contributed to rescue experiments. E.S. isolated and established the human mesenchymal stem cells from donors. P.M. provided software resources and performed analysis. P.M., N.G., and M.F. contributed to the project development, provided resources, and helped in manuscript writing. F.B. conceptualized and supervised the project and wrote the manuscript.

## Competing interests

The authors declare no competing interest.
