## [Peer Review File · Communications Biology]

Reviewers' comments:

Reviewer #1 (Remarks to the Author):

In this study, Blasi and colleagues report a novel role of PREP1 (PKNOX1), a transcription factor in regulating the expression of the proteins associated with the LINC complex such as SUN1, SUN2 and LAP2 as well as a role in regulating actomyosin contractility and focal adhesion morphology and cell traction. Additionally, the authors show a role for PREP1 in YAP localisation in the nucleus associated with mechanical cues from the extracellular environment. This study clearly shows that down-regulation of PREP1 changes the relative stoichiometry of the different LINC complex proteins which affects the stiffness of the nuclei. In addition, the phenotypes associated with focal adhesion and traction are also very clear. However, the connection between the nuclear phenotype, the adhesion and actin phenotype is unclear as is whether any of the nuclear phenotypes is because of the changes in expression level of SUN1 or 2 when PREP1 is down-regulated. Finally, the YAP results are contradictory (as stated by the authors) which leaves one a little confused but also intrigued. Overall, the authors should soften some of their conclusions, perhaps re-organize their results and if possible, include some rescue experiments for publication.

Specific comments:

1. The HeLa and U2OS results together are confusing especially because sometimes the results contradict each other. I would put all the results from one of the cell types (probably U2OS) in the main figure and move the HELA to supp figures (except perhaps fig 1). Additionally, at many places one phenotype is reported for one cell type and then another for the other. For eg. The focal adhesion area in Fig 4 are for U2OS but traction measurements for HeLa. It should either be for both or for one. Different cell types have different base levels of contractility and it may be that that is the diff between U2OS and HeLa. So, it would be good to see the YAP results with both the cell types too.
2. Data such as in Figure 3, si SUN1 (percentage deformed nuclei) is missing from the plot.
3. Since the depletion of SUN1 does not rescue the phenotype in the nucleus for PREP1 depletion, the value of figure 2 in the main text is unclear since there is no direct evidence of PREP1 mediated transcription regulating the functions described here.
4. The authors do a siSUN2 experiment, but have they tried an overexpression experiment with SUN2 on a siPREP1 background to see if that rescues the phenotype?
5. It will be good to see an siPREP1+PREP1 rescue experiment to make sure that the phenotypes observed are associated with loss of PREP1 directly.
6. Overall, the idea of nuclear stoichiometry is very interesting but unless the authors can show that the direct perturbation of this stoichiometry also affects focal adhesions, traction and apical actin, then the connection to PREP1 is very speculative and needs to be toned down.

Reviewer #2 (Remarks to the Author):

In the manuscript "Transcriptional regulation of cellular mechanosignaling by PREP1". This paper explores the role of transcription factor PREP1 (PKNOX1) in nuclear mechanics, shape, transcription, mechanotransduction, and the cytoskeleton. The paper reports some would be interesting results, but does so without sufficient rigorous experiments or data to support its major claims. The paper is built on the conclusion that PREP1 regulates SUN1, SUN2, and LAP2beta which in turn determine nuclear mechanics, morphology, and mechanotransduction. Evidence supporting this claim is not existent in my opinion. Thus, I suggest that this paper not be published until major revisions address that actual cause of nuclear mechanics and morphology, and how that or other factors disrupt YAP-TAZ mechanotransduction.

1) The manuscript does not provide a clear understanding of changes in nuclear mechanics and shape. In the abstract the sentence "PREP1 depletion upsets the nuclear membrane protein stoichiometry rendering nuclei soft." This statement is not supported by data. More importantly, the paper cannot determine the actual source of softer nucleus, which past literature has shown is the likely the cause of abnormal shape. Thus, the paper does not provide data to conclusively support two of the major phenotypes of decreased mechanics and shape. Instead the paper, here in the abstract and in many places, makes vague claims unsupported by data. This is a major flaw of the paper that renders it unpublishable in my opinion.

a. "PREP1 depletion leads to decrease in SUN2 and concomitant increase in SUN1 and LAP2 resulting in softer nuclei prone to deformation." (Page 2 line 63-63) This sentence in the last paragraph of the introduction has no data supporting it. The data does support SUN2 decrease and an increase in SUN1 and LAP2beta (but not alpha; Figure 1 and S1). However, this is a mixed bag in which depletion of SUN2 does not cause a significant increase in abnormal nuclear morphology (Figure 3B), and depletion of SUN1 (not increase as shown in PREP1 depletion) causes abnormal nuclear morphology (Figure 3A). The contribution of LAP2beta is not investigate and does not even discuss how it might contribute, even though the paper makes claims that it important. Thus the paper does not provide any such data that the decrease in nuclear mechanics and shape are understood in any way. Why does the paper attempt to make these unsupported conclusions?

i. "... (Fig. S1E), confirming that PREP1 affects nuclear mechanics by virtue of its influence on nuclear envelope proteins." (Page 4, line 114-115). This is another unsupported statement. The paper does not directly show/have data that the changes in nuclear envelope/lamina proteins directly result in decreased nuclear mechanics. The paper only measure decreased nuclear mechanics in U2OS and HELA cell lines wildtype vs siPREP1. The paper does not measure nuclear mechanics in any other nuclear envelope/lamin perturbation. Thus, the paper to be publishable must show what is the cause of decreased nuclear mechanics. This means nuclear mechanics measurements should be done upon disrupting the envelope proteins of interest in this study. This unsupported conclusion is repeated at the end of this paragraph.

ii. "PREP1 is able to regulate nuclear mechanics in a variety of cells through transcriptional regulation of SUN1, SUN2 and TMPO (LAP2) genes." (Page 5, 155-157) While the data does support a role for PREP1 in regulating these genes via binding in the genome, this data AGAIN in no way supports the changes in SUN1 and 2 and LAP2beta determine nuclear mechanics and also morphology. This continual use of this unsupported statement shows the paper is built on unsupported claims.

iii. The paper admits that these previous statements/conclusions to not explain abnormal nuclear morphology in PREP1 depletion. " All these data together with the fact that nuclear deformation was manifested only in U2OS cells, suggested that other factors, possibly nuclear extrinsic, are involved in the defective nuclear phenotype observed in PREP1 depleted cells." (page 6 167-169). Thus, the paper itself contradicts one of its most important conclusions. This section goes on to investigate if cytoskeletal factors contribute. The data here shows that decreasing actin compression rescues nuclear shape, this is not a novel finding as almost all perturbations of nuclear components resulting in abnormal nuclear shape are rescued upon disrupting actin and its compression.

iv. The list of citations provided in the manuscript are contradictory to the data presented here and/or do not support the conclusions of this paper. Thus, not only does the manuscript make conclusions unsupported by their data, but it also cites other papers that have data that is in direct conflict with the conclusions. The citations listed in the manuscript are listed below in bold and explained in italics.

19. Rowat, A. C., Lammerding, J. & Ipsen, J. H. Mechanical properties of the cell nucleus and the effect of emerin deficiency. *Biophys J* 91, 4649–4664 (2006). This is a relevant citation but not necessarily a relevant protein to the manuscript since Emerin levels were not shown to change by PREP1 depletion. Citation 12 is similar to this citation, investigating the protein Emerin.

20. Matsumoto, A. et al. Loss of the integral nuclear envelope protein SUN1 induces alteration of nucleoli. *Nucleus* 7, 68–83 (2016). This paper discusses nucleoli morphology not nucleus morphology. The effect is also opposite, cited paper SUN1 decrease and this manuscript presents SUN1 increase. Inclusion of this citation, I would argue, shows a lack of knowledge of the field of nuclear mechanics and morphology

21. Donahue, D. A. et al. SUN2 Overexpression Deforms Nuclear Shape and Inhibits HIV. *J Virol* 90, 4199–4214 (2016). This data is contradictory to that presented in the paper as siPREP1 decreases SUN2.

22. Chen, C.-Y. et al. Accumulation of the inner nuclear envelope protein Sun1 is pathogenic in progeric and dystrophic laminopathies. *Cell* 149, 565–77 (2012). This paper shows that LAMN-/- or D9 mutant over accumulate SUN1, decreasing this over accumulation rescues nuclear shape in these lamin A mutants. The cited paper is contradictory or does not explain the data in this manuscript. This paper sees an increase in SUN1 vis siPREP1, but dual siPREP1 and siSUN1 does not rescue deformed nuclei and IN FACT siSUN1 causes nuclear deformation to the by itself.

b. Furthermore, example images shown in Figure 3A of siPREP1 and siSUN1 show drastically abnormally shaped nuclei, where in Figure 3B the example images show 10 to 20 nuclei for control

and siSUN2 that appear to have a clear elliptical shape but still report 15% deformed nuclei. Why is there this clear visual difference? Can the paper display the levels of abnormal nuclear morphology differences by instead clearly reporting the shape characteristic of the nuclei – such as circularity, solidity, or other measurement that measure nucleus shape instead of a binary normal vs. deformed output?

i. Specifically, there is no mention of how a “deformed nucleus” is reported/measured. Is this a qualitative measure? The paper should have a materials and methods section clearly stating how they measured this reported data.

ii. In fact decreased SUN2 has been recently associated with decoupling for the actin cytoskeleton. Decoupling would be more akin to disrupting the actin cytoskeleton and relieving antagonistic forces on the nucleus that result in less deformed nuclei, not more nuclear deformation as the manuscript concludes.

c. Why is the data for lamin A mixed and unclear? The paper reports loss of lamin A via IF in Figure 1D. Oppositely, Western Blot data and transcript data shows no change in lamin A levels Figure S1 and 2. How does the paper reconcile this difference? The paper falsely suggests that change in IF levels but no change in WB is due to less incorporation of lamin A into the lamina. This is wrong because WB and IF are predicated on the same relative labeling scheme where and antibody binds to lamin A whether it is lamina incorporated or not incorporated. The materials and methods state that total signal for the nucleus (marked by Hoechst) was measured, is this true? If it is then free floating Lamin A will provide IF signal in the middle of the nucleus, meaning total levels of Lamin A measured in this way should be the same as Western Blot, which they are not. It is important that the paper deal with this conflicting data. If the paper wants to make a claim about incorporated vs free lamin A then they should measure that in a more direct way. It is well known that phosphorylated Lamin A (P22) is not incorporated in the lamina (Kochin et al 2014). The paper should measure these specific pools of lamin A to support the currently unsupported conclusions in the paper.

i. “However, we detected lower LAMIN B1 intensity in PREP1 depleted HeLa cells, indicating that changes in nuclear lamins is probably a secondary phenotype (Fig. S1D).” (Page 4 111-112) This is conflicting logic. Both lamin B1 levels and nuclear morphology wild-type vs siPREP1 are different between U2OS and HELA cells. However, for Lamin B1 the conclusion is that it is a “secondary phenotype” while for nuclear shape the paper works to understand why instead of labeling it as a secondary phenotype. How can the authors justify picking and choosing between these two for similar instances as important vs unimportant (secondary phenotype)?

d. “Further, we propose that nucleo-cytoskeletal coupling by PREP1 is necessary for the correct mechanotransduction of YAP-TAZ.” (Page 9, 275) While an interesting idea, this final conclusion of the results does not consider that the softer nucleus could be the result of disrupted mechanotransduction YAP-TAZ signaling. The manuscript does not attempt to disrupt the nucleo-cytoskeletal components (SUN1,2 and LAPbeta) that are disrupted in PREP1 depletion. In fact, the paper shows the capability to disrupt SUN1 and 2, but they were not tested to determine if their diction had any effect on YAP-TAZ. Thus this conclusion, as most in the paper, lacks rigorous experimental data to support it.

e. Finally, the paper does not once address the role of chromatin and chromatin proteins in nuclear mechanics and shape. The contribution of chromatin proteins was first established in 2013 with Imbalzano et al PLOS One and followed by Furusawa 2015 Nat Comm. The past 7 years have brought many more publications showing that chromatin and chromatin proteins are major contributors the nuclear mechanics and shape. The fact that a paper on a chromatin protein PREP1 does not acknowledge this fact suggest a gross lack knowledge leading to the paper not considering key experimental parameters.

Reviewer #3 (Remarks to the Author):

In this work, Purushothaman et al show How PREP1 regulates expression of nuclear membrane and LINC complex proteins. Authors show how alterations of nuclear membrane affect intermediate filaments LAMIN A/C and LAMIN B and how this changes the mechanical properties of nuclei. Finally, they show how PREP1 affects nuclear localization of transcription regulator YAP. In my view the paper shows novel and relevant results and the text is well written. However, there are some aspects as data quantification and some clarifications that authors need to address.

Specific comments below:

Major

1- Authors make many claims from images and western blots without quantifying data (Fig 1E, 1F, 3A, 3B, 3C, 3D, 3E...). All immunoblots and immunofluorescence images need to be quantified before making any conclusions.

2- In a similar way, the deformations of the nuclei produced by different siRNA can give more information than what obtained by counting deformed nuclei. Please set some specific rules to quantify nuclear deformation that can bring more information and can also be more reproducible.

3- Fig 3F shows a restoration to a certain extent in % of deformed nuclei, however the restoration is not complete. This should be pointed out. Again, % of deformed nuclei is not the best way to quantify the effect on nuclear shape, as pointed out before. Furthermore, as actomyosin contraction is affected, authors should measure the nucleus shape in 3D and compare to control to see if there is restoration

4- I am unsure about the measurements of p-YAP and YAP levels. Is the YAP antibody unable to recognize p-YAP? p-YAP levels seem to be higher than YAP ones at the Western blots of fig 5-C. It has also been shown that p-YAP can enter the nucleus with substrate rigidity and that not all non p-YAP is nuclear. I think that performing immunostainings of YAP (done in fig5A) and p-YAP could be a clearer way to measure how PERP1 depletion is affecting YAP localization.

Minor

1- Authors see a decrease of LAMIN A/C levels by immunofluorescence mean grey values (Fig 1D). However, they do not observe any change in LAMIN A/C expression by immunoblot and no different transcription levels by q-PCR. Could the authors discuss this point?

2- Authors claim that under PREP1 depletion nuclei are "softer and prone to deformation". However, nuclei deformations only occur in U2OS cell, but not in HeLa cells. Please state that nuclear deformation is cell type dependent.

3- What is the scale [X Y] of traction maps? They also lack a scale bar.

4- Images of figure 5 A do not seem to be representative of the quantification, as one expects not to be able to define the nuclei of cells through YAP images in siPREP1 cells (ratio close to 1), but this is not the case.

Dear Editor,

We are submitting the revised manuscript "Transcriptional Regulation of Cellular Mechanosignaling by PREP1 (PKNOX1)" by Purushothaman et.al. The revision took more time than we anticipated due to technical difficulties and the extreme circumstances created by the pandemic. We take this opportunity to thank you for waiving us any timelimit on the submission. We have gone through the reviewers' comments and your suggestions and have added more experiments which we believe enhanced the scientific rigor of the paper. Below is the summary of your suggestions and how we addressed them. In this rebuttal, we have included a point-by-point reply to the reviewers' comments. We thank the referees for their valuable suggestions and have included more experiments which would address the concerns of the referees. We also explain why certain suggestions, while we welcome, are out of the scope of this manuscript.

Summary comments from the editor

In particular, please note that the following revisions would be necessary for us to contact our referees again:

1. Further characterize the mechanism of PREP1 by performing rescue experiments as requested by Reviewers #1 and #2 "for example Reviewer #1 said: It will be good to see an siPREP1+PREP1 rescue experiment to make sure that the phenotypes observed are associated with loss of PREP1 directly.". We appreciate this suggestion. We have included rescue experiments in the revised manuscript

2. Provide more experimental evidence to support that PREP1 regulates SUN1, SUN2, and LAP2beta which in turn determine nuclear mechanics, morphology, and mechanotransduction as pointed by Reviewer #2.

The fact that PREP1 and its interactors are transcriptional regulators is an established fact. While we believe that intricate mechanism by which SUN1, SUN2 and LAP2beta is regulated by PREP1 is interesting, we think such an exercise is beyond the scope of this study and does not influence the main message of the manuscript. What we show in the manuscript is that PREP1, a transcription factor, can regulate the expression of SUN1, SUN2 and LAP2. We have provided protein and mRNA expression data on PREP1 depleted cells together with ChIPseq data of PREP1 binding in these target genes. We do not comment whether this transcriptional regulation is direct or indirect in the paper. Unraveling the mechanism is not a single experiment but a project in itself. Whether it is direct or indirect, we have added rescue experiments to address the specificity of this regulation.

The main message of the paper is that transcription factor PREP1 can regulate transcript levels of SUN1, SUN2 and LAP2 (directly or indirectly) which makes the nuclei softer. However, the reduction in nuclear stiffness is not accompanied by a reduction in cellular stiffness since actin cytoskeleton and focal adhesions are enriched and cells exert higher traction on the substratum. Surprisingly YAP-TAZ is not translocated to the nucleus in this situation following the conventional wisdom (higher the traction, focal adhesions and actin cytoskeleton, higher translocation of YAP to the nucleus). Hence, we propose that PREP1 is necessary for co-ordinated nucleo-cytoskeleton.

3. Provide quantification for all immunoblots and immunofluorescence data as requested by Reviewer #3.

We have provided quantification for IF and IF data. Proper quantification of nuclear phenotype also will be included as suggested by reviewer 3.

4. Comprehensively address all the other concerns raised by the referees. Many aspects regarding data quantification and clarifications of results are needed.

We have addressed all the concerns raised by the referees point-by point in the response to referees.

We thank you for your valuable help and hope that we have adequately addressed the concerns of the referees

Individual Referee comments and Rebuttal

Referee 1

Specific comments:

1. The HeLa and U2OS results together are confusing especially because sometimes the results contradict each other. I would put all the results from one of the cell types (probably U2OS) in the main figure and move the HELA to supp figures (except perhaps fig 1). Additionally, at many places one phenotype is reported for one cell type and then another for the other. For eg. The focal adhesion area in Fig 4 are for U2OS but traction measurements for HeLa. It should either be for both or for one. Different cell types have different base levels of contractility and it may be that that is the diff between U2OS and HeLa. So, it would be good to see the YAP results with both the cell types too.

We thank the reviewer for their suggestion. We have reorganized the manuscript by keeping U2OS cell line results in the main figure and HeLa in the supplementary section. We have also provided additional data on U2OS cell line which were previously lacking (traction force microscopy) making it more streamlined.

2. Data such as in Figure 3, si SUN1 (percentage deformed nuclei) is missing from the plot.

We thank the reviewer for bringing this oversight. The corrected graph is added to the figure.

3. Since the depletion of SUN1 does not rescue the phenotype in the nucleus for PREP1 depletion, the value of figure 2 in the main text is unclear since there is no direct evidence of PREP1 mediated transcription regulating the functions described here.

We believe this comment is due to the way we structured our manuscript and we thank the reviewer for bringing it to our attention. We have now explained it clearly in the manuscript that downregulation of PREP1 makes the nuclei softer and upregulate SUN1 and LAP2b and downregulate SUN2, in both HeLa and U2OS cells. This is the nuclear phenotype arising from PREP1 downregulation which we have shown is due to the transcriptional regulation. We have included data which shows that depletion of SUN2 alone could also cause softer nuclei. Nuclear deformation which we see in U2OS cells alone is cell type specific presentation of the nuclear defects of PREP1 due to extracellular cytosolic forces specific to that cell type. Hence, even if SUN1 depletion did not rescue the nuclear deformation in U2OS cells, the fact that PREP1 can transcriptionally regulate these proteins is relevant in our opinion. We have now included rescue experiments like the reviewer suggested which will further address this issue.

4. The authors do a siSUN2 experiment, but have they tried an overexpression experiment with SUN2 on a siPREP1 background to see if that rescues the phenotype?

Yes, we have performed these experiments. However, the viability of cells after the overexpression preclude any meaningful analysis. Further, we could not see any difference in the nuclear shape of cells expressing SUN2 vs cells that do not.

5. It will be good to see an siPREP1+PREP1 rescue experiment to make sure that the phenotypes observed are associated with loss of PREP1 directly.

We thank the reviewer for this suggestion. We have tried to do the rescue experiments in U2OS cells; however due to technical difficulties explained in the manuscript, we had used another cell line with no or little expression of PREP1 to address these issues. We hope the reviewer will find these results address his concerns adequately.

Overall, the idea of nuclear stoichiometry is very interesting but unless the authors can show that the direct perturbation of this stoichiometry also affects focal adhesions, traction and apical actin, then the connection to PREP1 is very speculative and needs to be toned down.

We agree with the reviewer and will tone down the speculations

Reviewer 2

1) The manuscript does not provide a clear understanding of changes in nuclear mechanics and shape. In the abstract the sentence "PREP1 depletion upsets the nuclear membrane protein stoichiometry rendering nuclei soft." This statement is not supported by data. More importantly, the paper cannot determine the actual source of softer nucleus, which past literature has shown is likely the cause of abnormal shape. Thus, the paper does not provide data to conclusively support two of the major phenotypes of decreased mechanics and shape. Instead the paper, here in the abstract and in many places, makes vague claims unsupported by data. This is a major flaw of the paper that renders it unpublishable in my opinion.

We have provided data AFM data on U2OS and Hela cells (Fig 1 and Fig. S1) which showed that Prep1 depletion causes softer nuclei. The same cells also show changes in nuclear membrane proteins SUN1, SUN2 and LAP2. Studies have shown that disruption of lamina and associated proteins can perturb cellular and nuclear stiffness (Lammerding et.al., JCI, 2004, Stewart Hutchinson et.al., Exp. Cell research 2008, Liu et.al., Biochem and Cell Biol. 2019). In the revised manuscript, we have included the AFM analysis of siSUN2 cells which show softer nuclei. In Hela cells, isolated nuclei itself is softer, hence we propose that nuclear softness is due to the perturbation in nuclear membrane proteins rather than due to extra nuclear forces. Further, abnormal nuclear shape is observed in softer and stiffer nuclei for example; HGPS cells(Dahl et al., 2006; Verstraeten et al., 2008). Therefore, equating nuclear softness with nuclear deformation is not informative and in fact misleading. In this paper, we have made this distinction clear with the use of two cell lines (Hela and U2OS which shows similar nuclear softness but different nuclear morphology). Hence, we assessed the involvement perinuclear actin cytoskeleton in modulating nuclear morphology in U2OS cells. We believe the reviewer has overlooked this fact in making his assessment.

a. "PREP1 depletion leads to decrease in SUN2 and concomitant increase in SUN1 and LAP2 resulting in softer nuclei prone to deformation." (Page 2 line 63-63) This sentence in the last paragraph of the introduction has no data supporting it. The data does support SUN2 decrease and an increase in SUN1 and LAP2beta (but not alpha; Figure 1 and S1). However, this is a mixed bag in which depletion of SUN2 does not cause a significant increase in abnormal nuclear morphology (Figure 3B), and depletion of SUN1 (not increase as shown in PREP1 depletion) causes abnormal nuclear morphology (Figure 3A). The contribution of LAP2beta is not investigate and does not even discuss how it might contribute, even though the paper makes claims that it important. Thus the paper does not provide any such data that the decrease in nuclear mechanics and shape are understood in any way. Why does the paper attempt to make these unsupported conclusions?

We believe that this criticism is again due to equating nuclear softness with nuclear deformation. As we pointed out above, perturbation of nuclear lamina proteins could affect the nuclear stiffness. However, nuclear deformation is due to increased perinuclear actin formation which is clearly shown in Figure 4 of the paper. The reviewer completely overlooks this data or the explanation we have

given in the text and repeats the same point. We agree that we did not make investigate LAP2beta in detail since it was out of the scope of this manuscript.

i. "... (Fig. S1E), confirming that PREP1 affects nuclear mechanics by virtue of its influence on nuclear envelope proteins." (Page 4, line 114-115). This is another unsupported statement. The paper does not directly show/have data that the changes in nuclear envelope/lamina proteins directly result in decreased nuclear mechanics. The paper only measure decreased nuclear mechanics in U2OS and HELA cell lines wildtype vs siPREP1. The paper does not measure nuclear mechanics in any other nuclear envelope/lamin perturbation. Thus, the paper to be publishable must show what is the cause of decreased nuclear mechanics. This means nuclear mechanics measurements should be done upon disrupting the envelope proteins of interest in this study. This unsupported conclusion is repeated at the end of this paragraph.

We disagree with the reviewer on this solely because our aim in the paper is not to claim that changes in nuclear envelope/lamina proteins **directly** result in decreased nuclear mechanics. Previous studies have given ample evidences on how disruption of lamins or lamina associated proteins can affect nuclear mechanics (either directly or indirectly through changing chromatin architecture). Further, emulating the exact composition of nuclear envelope upon PREP1 depletion is near to impossible. However, we already have data on SUN2 depleted cells which shows softer nuclei and is included in the manuscript.

ii. "PREP1 is able to regulate nuclear mechanics in a variety of cells through transcriptional regulation of SUN1, SUN2 and TMPO (LAP2) genes." (Page 5, 155-157) While the data does support a role for PREP1 in regulating these genes via binding in the genome, this data AGAIN in no way supports the changes in SUN1 and 2 and LAP2beta determine nuclear mechanics and also morphology. This continual use of this unsupported statement shows the paper is built on unsupported claims.

We would like to repeat that changes in nuclear membrane proteins only changes the nuclear stiffness but not morphology. We have clearly shown that morphology is regulated by extranuclear factors (Figure 4). As explained above, we have not individually tested the effect of these proteins on nuclear softness since loss of lamin A (similar to what happens in PREP1 depletion) causes changes in nuclear stiffness. Therefore, we are unable to understand the reviewer's criticism here. We never claimed that individual proteins are important. We have always maintained that the effect is complex and nuanced. We have rewritten the manuscript and try to convey our message correctly, in case the reviewer is misled by it

iii. The paper admits that these previous statements/conclusions do not explain abnormal nuclear morphology in PREP1 depletion. " All these data together with the fact that nuclear deformation was manifested only in U2OS cells, suggested that other factors, possibly nuclear extrinsic, are involved in the defective nuclear phenotype observed in PREP1 depleted cells." (page 6 167-169). Thus, the paper itself contradicts one of its most important conclusions. This section goes on to investigate if cytoskeletal factors contribute. The data here shows that decreasing actin compression rescues nuclear shape, this is not a novel finding as almost all perturbations of nuclear components resulting in abnormal nuclear shape are rescued upon disrupting actin and its compression.

We believe this criticism again rises from the fact that reviewer is reading nuclear softness equal to nuclear deformation. We have structured the paper starting with nuclear defects, investigating the changes in nuclear proteins building up to nuclear extrinsic factors since that was the logical path. Our experiments with two cell lines (Fig 1) and single depletions of SUN2, SUN1 (Fig. 3) suggested that nuclear softness does not always lead to nuclear deformation, hence we moved onto cytoskeleton and characterized perinuclear actin formation in Hela and U2OS cells. We do not claim that it is a novel finding as we have cited papers which made those findings.

iv. The list of citations provided in the manuscript are contradictory to the data presented here and/or do not support the conclusions of this paper. Thus, not only does the manuscript make conclusions unsupported by their data, but it also cites other papers that have data that is in direct conflict with the conclusions. The citations listed in the manuscript are listed below in bold and explained in italics.

19. Rowat, A. C., Lammerding, J. & Ipsen, J. H. Mechanical properties of the cell nucleus and the effect of emerin deficiency. *Biophys J* 91, 4649–4664 (2006). This is a relevant citation but not necessarily a relevant protein to the manuscript since Emerin levels were not shown to change by PREP1 depletion. Citation 12 is similar to this citation, investigating the protein Emerin.

20. Matsumoto, A. et al. Loss of the integral nuclear envelope protein SUN1 induces alteration of nucleoli. *Nucleus* 7, 68–83 (2016). This paper discusses nucleoli morphology not nucleus morphology. The effect is also opposite, cited paper SUN1 decrease and this manuscript presents SUN1 increase. Inclusion of this citation, I would argue, shows a lack of knowledge of the field of nuclear mechanics and morphology

21. Donahue, D. A. et al. SUN2 Overexpression Deforms Nuclear Shape and Inhibits HIV. *J Virol* 90, 4199–4214 (2016). This data is contradictory to that presented in the paper as siPREP1 decreases SUN2.

22. Chen, C.-Y. et al. Accumulation of the inner nuclear envelope protein Sun1 is pathogenic in progeric and dystrophic laminopathies. *Cell* 149, 565–77 (2012). This paper shows that LAMN-/- or D9 mutant over accumulate SUN1, decreasing this over accumulation rescues nuclear shape in these lamin A mutants. The cited paper is contradictory or does not explain the data in this manuscript. This paper sees an increase in SUN1 vis siPREP1, but dual siPREP1 and siSUN1 does not rescue deformed nuclei and IN FACT siSUN1 causes nuclear deformation to the by itself.

The citations which the reviewer mentioned were not cited as support for our specific claims but as studies which suggest general perturbations in nuclear membrane proteins can affect nuclear mechanics. Reference 22 was cited as a pretext to test whether SUN1 depletion would rescue the phenotype as one would assume as per the paper cited, but as we clearly mentioned in the manuscript it doesn't. We did not claim that this paper explains the data.

However, the citation 20 was an oversight which would have gone unnoticed and we thank the reviewer for the detailed check.

b. Furthermore, example images shown in Figure 3A of siPREP1 and siSUN1 show drastically abnormally shaped nuclei, where in Figure 3B the example images show 10 to 20 nuclei for control and siSUN2 that appear to have a clear elliptical shape but still report 15% deformed nuclei. Why is there this clear visual difference? Can the paper display the levels of abnormal nuclear morphology differences by instead clearly reporting the shape characteristic of the nuclei – such as circularity, solidity, or other measurement that measure nucleus shape instead of a binary normal vs. deformed output?

i. Specifically, there is no mention of how a “deformed nucleus” is reported/measured. Is this a qualitative measure? The paper should have a materials and methods section clearly stating how they measured this reported data.

We thank the reviewer for this suggestion. In the revised manuscript, we have included graphs (Supplementary Fig. 1) with nuclear shape parameters description and quantitation.

ii. In fact decreased SUN2 has been recently associated with decoupling for the actin cytoskeleton. Decoupling would be more akin to disrupting the actin cytoskeleton and relieving antagonistic forces on the nucleus that result in less deformed nuclei, not more nuclear deformation as the manuscript concludes.

We agree with the reviewer that the changes in SUN2 should decouple and relieve the antagonistic forces on the nuclei. However, there is an increase in SUN1 which could compensate for the loss as the detailed stoichiometric interactions between SUN1/SUN2 and NESPRINS is not very clear.

c. Why is the data for lamin A mixed and unclear? The paper reports loss of lamin A via IF in Figure 1D. Oppositely, Western Blot data and transcript data shows no change in lamin A levels Figure S1 and 2. How does the paper reconcile this difference? The paper falsely suggests that change in IF levels but no change in WB is due to less incorporation of lamin A into the lamina. This is wrong because WB and IF are predicated on the same relative labeling scheme where an antibody binds to lamin A whether it is lamina incorporated or not incorporated. The materials and methods state that total signal for the nucleus (marked by Hoechst) was measured, is this true? If it is then free floating Lamin A will provide IF signal in the middle of the nucleus, meaning total levels of Lamin A measured in this way should be the same as Western Blot, which they are not. It is important that the paper deal with this conflicting data. If the paper wants to make a claim about incorporated vs free lamin A then they should measure that in a more direct way. It is well known that phosphorylated Lamin A (P22) is not incorporated in the lamina (Kochin et al 2014). The paper should measure these specific pools of lamin A to support the currently unsupported conclusions in the paper.

We agree with the reviewer that WB and IF predicated on the same relative labeling scheme. However, they are two different techniques and to compare the results would be entirely inappropriate.

Firstly, IF and imaging offers spatial resolution of the signal. Depending on the voltage and offset settings diffuse signals may not be acquired. It is not meant to be a quantitative method though it can be used so if desired. WB on the other hand is quantitative and gives the total protein level with much accuracy. Secondly, even though we use the same antibody in both techniques, in IF antigen is in native state while in WB, proteins are denatured unless we run native gels. Binding efficiency of antibodies can vary vastly depending on this. That being said, WB results also depend on the solubility of the proteins. Lamins are relatively insoluble proteins and this could affect the results. Finally, there are published reports which suggests epitope masking of Lamin A at the basal level which is dependent on the polarity, LINC complex and actin cytoskeleton (Ihalainen et.al., 2015). We have included this in the discussion of the revised manuscript

i. "However, we detected lower LAMIN B1 intensity in PREP1 depleted HeLa cells, indicating that changes in nuclear lamins is probably a secondary phenotype (Fig. S1D)." (Page 4 111-112) This is conflicting logic. Both lamin B1 levels and nuclear morphology wild-type vs siPREP1 are different between U2OS and HELA cells. However, for Lamin B1 the conclusion is that it is a "secondary phenotype" while for nuclear shape the paper works to understand why instead of labeling it as a secondary phenotype. How can the authors justify picking and choosing between these two for similar instances as important vs unimportant (secondary phenotype)?

We would like to clarify here that we believe changes in **nuclear lamins** to be a secondary phenotype not specifically Lamin B as understood by the reviewer. We apologize that the quoted sentence is particularly misleading. We made this conclusion since no change in mRNA levels were observed in either Lamin A or Lamin B but we have moved the sentence to later sections where mRNA levels are shown.

d. "Further, we propose that nucleo-cytoskeletal coupling by PREP1 is necessary for the correct mechanotransduction of YAP-TAZ." (Page 9, 275) While an interesting idea, this final conclusion of the results does not consider that the softer nucleus could be the result of disrupted mechanotransduction YAP-TAZ signaling. The manuscript does not attempt to disrupt the nucleo-cytoskeletal components (SUN1,2 and LAPbeta) that are disrupted in PREP1 depletion. In fact, the paper shows the capability to disrupt SUN1 and 2, but they were not tested to determine if their

diction had any effect on YAP-TAZ. Thus this conclusion, as most in the paper, lacks rigorous experimental data to support it.

In the manuscript, we have measured nuclear translocation of YAP to understand if the increased FA pathway signaling would promote its nuclear localization despite a softer nucleus, Not to pinpoint the reason for nuclear softness. We thank the reviewer for the suggestion to check YAP-TAZ localization in siSUN1 and siSUN2 and have included it in the revised manuscript.

e. Finally, the paper does not once address the role of chromatin and chromatin proteins in nuclear mechanics and shape. The contribution of chromatin proteins was first established in 2013 with Imbalzano et al PLOS One and followed by Furusawa 2015 Nat Comm. The past 7 years have brought many more publications showing that chromatin and chromatin proteins are major contributors the nuclear mechanics and shape. The fact that a paper on a chromatin protein PREP1 does not acknowledge this fact suggest a gross lack knowledge leading to the paper not considering key experimental parameters.

We agree with the reviewer that chromatin and various chromatin proteins are involved in the regulation of nuclear mechanics and shape. Interfering with a chromatin binding protein like PREP1 is bound to have direct and indirect consequences on many chromatin proteins. Hence, we have pursued an approach which is more in line with its proven role as a transcriptional regulator and we showed in the manuscript that there is change at the transcriptional regulation of inner nuclear membrane proteins SUN1 and SUN2 in PREP1 depleted cells. We have also looked at chromatin binding protein BANF1 which acts as a bridge between chromatin and lamin associated proteins and we did not see any change there. Histone levels were also not changed. While we do not know the protein status, in the first glance, HMGN5 is reduced while no change was detected in BRG1 in the RNA-seq analysis of PREP1 depleted cells. As per published studies, reduced HMGN5 should be associated with increased chromatin compaction and sturdy nucleus as opposed to what is observed PREP1 depletion.

Reviewer 3

Major

1- Authors make many claims from images and western blots without quantifying data (Fig 1E, 1F, 3A, 3B, 3C, 3D, 3E...). All immunoblots and immunofluorescence images need to be quantified before making any conclusions.

We thank the reviewer for this comment and have provided quantifications for the immunoblots in the revised version.

2- In a similar way, the deformations of the nuclei produced by different siRNA can give more information than what obtained by counting deformed nuclei. Please set some specific rules to quantify nuclear deformation that can bring more information and can also be more reproducible.

We thank the reviewer for this suggestion and have included specific parameter descriptions in the revised manuscript which addresses this comment.

3- Fig 3F shows a restoration to a certain extent in % of deformed nuclei, however the restoration is not complete. This should be pointed out. Again, % of deformed nuclei is not the best way to quantify the effect on nuclear shape, as pointed out before. Furthermore, as actomyosin contraction is affected, authors should measure the nucleus shape in 3D and compare to control to see if there is restoration

We thank the reviewer for this suggestion. We agree with the reviewer that the rescue is not 100% and have included restoration of nuclear parameters in the revised manuscript.

4- I am unsure about the measurements of p-YAP and YAP levels. Is the YAP antibody unable to recognize p-YAP? p-YAP levels seem to be higher than YAP ones at the Wester blots of fig 5-C. It has also been shown that p-YAP can enter the nucleus with substrate rigidity and that not all non p-YAP

is nuclear. I think that performing immunostainings of YAP (done in fig5A) and p-YAP could be a clearer way to measure how PERP1 depletion is affecting YAP localization.

We agree with the reviewer about the discrepancy between p-YAP and YAP antibody which could simply be due to the binding efficiency of the respective antibodies. However, this is just a speculation. We have done immunostainings of YAP in different substrate rigidity conditions and quantified the nuclear/Cytoplasmic ratio as the reviewer suggested. This is included in the revised manuscript.

Minor

1- Authors see a decrease of LAMIN A/C levels by immunofluorescence mean grey values (Fig 1D). However, they do not observe any change in LAMIN A/C expression by immunoblot and no different transcription levels by q-PCR. Could the authors discuss this point?

We have included this in the revised manuscript.

2- Authors claim that under PERP1 depletion nuclei are “softer and prone to deformation”. However, nuclei deformations only occur in U2OS cell, but not in HeLa cells. Please state that nuclear deformation is cell type dependent.

We thank the reviewer for this comment. We have specified that deformation is cell type specific in the revised manuscript.

3- What is the scale [X Y] of traction maps? They also lack a scale bar.

We have relabeled the scales on traction maps as they were clearly illegible. We thank the reviewer for pointing it out.

4- Images of figure 5 A do not seem to be representative of the quantification, as one expects not to be able to define the nuclei of cells through YAP images in siPERP1 cells (ratio close to 1), but this is not the case.

We thank the reviewer for this comment. We have replaced it with a more representative image in the revised manuscript. siPERP1 cells obviously have cells with defined nuclei however there are more cells with dispersed YAP distribution.

REVIEWERS' COMMENTS:

Reviewer #1 (Remarks to the Author):

The revised manuscript and letter have addressed most of my concerns. I must commend the authors for trying to address all the points raised in the first round of reviews. The interpretation of data in these experiments are complicated and often confounding but I guess that's the beauty of cell biology. Every piece of a puzzle doesn't need to always fit perfectly but it's important to have these results out there!

My only 2 minor comments:

If possible, include the model/summary figure to help with interpretation of data (especially comparing U2OS and HeLa).

Include statistical repeats on some of the blots in the main figure.

Reviewer #2 (Remarks to the Author):

The paper has been sufficiently rewritten to communicate the data and the complexity of the PREP1 downstream events. All previous unsupported claims have been removed or clarified. The paper has had more data added to it to provide a better understanding of the results. This paper will be of general interest to the nuclear mechanobiology community and thus I believe should be published.

Reviewer #3 (Remarks to the Author):

Authors have addressed my concerns. I still miss the scale bar of the traction maps and scale bars of the immunostaining images are too small. After these minor corrections I recommend the manuscript for publication.